# BOOT🥾: Data-free Distillation of Denoising Diffusion Models with Bootstrapping

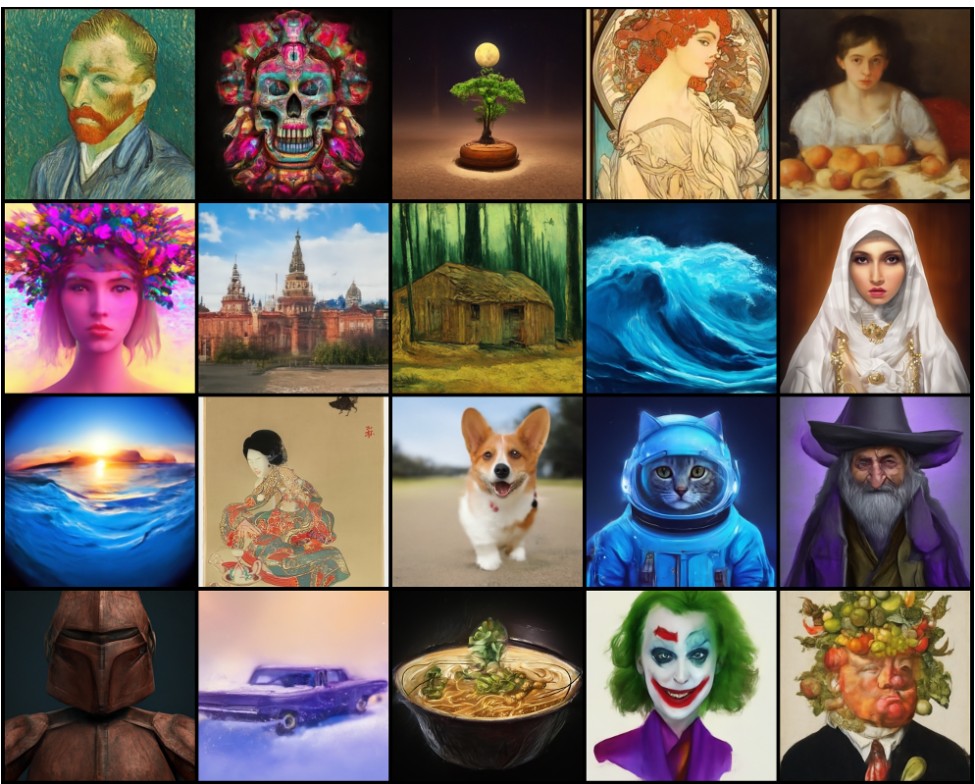

Figure 1: Samples of our distilled **single-step** model with prompts from *diffusiondb*.

## Abstract

Diffusion models have demonstrated great potential for generating diverse images. However, their performance often suffers from slow generation due to iterative denoising. Knowledge distillation has been recently proposed as a remedy which can reduce the number of inference steps to one or a few, without significant quality degradation. However, existing distillation methods either require significant amounts of offline computation for generating synthetic training data from the teacher model, or need to perform expensive online learning with the help of real data. In this work, we present a novel technique called *BOOT*, that overcomes these limitations with an efficient data-free distillation algorithm. The core idea is to learn a time-conditioned model that predicts the output of a pre-trained diffusion model teacher given any time-step. Such a model can be efficiently trained based on bootstrapping from two consecutive sampled steps. Furthermore, our method can be easily adapted to large-scale text-to-image diffusion models, which are challenging for previous methods given the fact that the training sets are often large and difficult to access. We demonstrate the effectiveness of our approach on several benchmark datasets in the DDIM setting, achieving comparable generation quality while being orders of magnitude faster than the diffusion teacher. The text-to-image results show that the proposed approach is able to handle highly complex distributions, shedding light on more efficient generative modeling.

# 1 INTRODUCTION

Diffusion models (Sohl-Dickstein et al., 2015; Ho et al., 2020; Nichol & Dhariwal, 2021; Song et al., 2020b) have become the standard tools for generative applications, such as image (Dhariwal & Nichol, 2021; Rombach et al., 2021; Ramesh et al., 2022; Saharia et al., 2022), video (Ho et al., 2022b;a), 3D (Poole et al., 2022; Gu et al., 2023; Liu et al., 2023b; Chen et al., 2023), audio (Liu et al., 2023a), and text (Li et al., 2022; Zhang et al., 2023) generation. Diffusion models are considered more stable for training compared to alternative approaches like GANs (Goodfellow et al., 2014a) or VAEs (Kingma & Welling, 2013), as they don't require balancing two modules, making them less susceptible to issues like mode collapse or posterior collapse. Despite their empirical success, standard diffusion models often have slow inference times (around $50 \sim 1000\times$ slower than single-step models like GANs), which poses challenges for deployment on consumer devices. This is mainly because diffusion models use an iterative refinement process to generate samples.

To address this issue, previous studies have proposed using knowledge distillation to improve the inference speed (Hinton et al., 2015). The idea is to train a faster student model that can replicate the output of a pre-trained diffusion model. In this work, we focus on learning efficient *single-step* models that only require one neural function evaluation (NFE). However, previous methods, such as Luhman & Luhman (2021), require executing the full teacher sampling to generate synthetic targets for every student update, which is impractical for distilling large diffusion models like StableDiffusion (SD, Rombach et al., 2021). Recently, several techniques have been proposed to avoid sampling using the concept of "bootstrap". For example, Salimans & Ho (2022) gradually reduces the number of inference steps based on the previous stage's student, while Song et al. (2023) and Berthelot et al. (2023) train single-step denoisers by enforcing self-consistency between adjacent student outputs along the same diffusion trajectory. However, these approaches rely on the availability of real data to simulate the intermediate diffusion states as input, which limits their applicability in scenarios where the desired real data is not accessible.

In this paper, we propose *BOOT*, a *data-free* knowledge distillation method for denoising diffusion models, with *single-step* inference via bootstrapping. Our inspiration for BOOT partially draws from the insight presented by consistency model (CM, Song et al., 2023) that all points on the same diffusion trajectory, *a.k.a.*, PF-ODE (Song et al., 2020b), have a deterministic mapping between each other. We identify two advantages of the proposed method: ($i$) Similar to CM, BOOT enjoys efficient *single-step* inference which dramatically facilitates the model deployment on scenarios demanding low resource/latency. ($ii$) Different from CM, which seeks self-consistency from any $x_t$ to $x_0$, thus being data-dependent, BOOT predicts all possible $x_t$ given the same noise point $\epsilon$ and a time indicator $t$. Consequently, BOOT $g_\theta$ always reads pure Gaussian noise, making it *data-free*. Moreover, learning all $x_t$ from the same $\epsilon$ enables bootstrapping: it is easier to predict $x_t$ if the model has already learned to generate $x_{t'}$ where $t' > t$. However, formulating bootstrapping in this way presents additional non-trivial challenges, such as noisy sample prediction. To address this, we learn the student model from a novel *Signal-ODE* derived from the original PF-ODE. We also design objectives and boundary conditions to enhance the sampling quality and diversity. This enables efficient inference of large diffusion models in scenarios where the original training corpus is inaccessible due to privacy or other concerns. For example, we can obtain an efficient model for synthesizing images of *"raccoon astronaut"* by distilling the text-to-image model with the corresponding prompts (shown in Fig. 2), even though collecting such real data is difficult.

In the experiments, we first demonstrate the efficacy of BOOT on various challenging image generation benchmarks, including unconditional and class-conditional settings. Next, we show that the proposed method can be easily adopted to distill text-to-image diffusion models. An illustration of sampled images from our distilled text-to-image model is shown in Fig. 1.

# 2 PRELIMINARIES

## 2.1 DIFFUSION MODELS

Diffusion models (Sohl-Dickstein et al., 2015; Song & Ermon, 2019; Ho et al., 2020) belong to a class of deep generative models that generate data by progressively removing noise from the initial input. In this work, we focus on continuous-time diffusion models (Song et al., 2020b; Kingma et al., 2021; Karras et al., 2022) in the variance-preserving formulation (Salimans & Ho, 2022). Given a

data point $\boldsymbol{x} \in \mathbb{R}^N$, we model a series of time-dependent latent variables $\{\boldsymbol{x}_t | t \in [0, T], \boldsymbol{x}_0 = \boldsymbol{x}\}$ based on a given noise schedule $\{\alpha_t, \sigma_t\}$:

$$q(\boldsymbol{x}_t | \boldsymbol{x}_s) = \mathcal{N}(\boldsymbol{x}_t; \alpha_{t|s} \boldsymbol{x}_s, \sigma_{t|s}^2 I), \text{ and } q(\boldsymbol{x}_t | \boldsymbol{x}) = \mathcal{N}(\boldsymbol{x}_t; \alpha_t \boldsymbol{x}, \sigma_t^2 I),$$

where $\alpha_{t|s} = \alpha_t / \alpha_s$ and $\sigma_{t|s}^2 = \sigma_t^2 - \alpha_{t|s}^2 \sigma_s^2$ for $s < t$. By default, the signal-to-noise ratio (SNR, $\alpha_t^2 / \sigma_t^2$) decreases monotonically with $t$. A diffusion model $\boldsymbol{f}_\phi$ learns to reverse the diffusion process by denoising $\boldsymbol{x}_t$. After training, one can use ancestral sampling (Ho et al., 2020) to synthesize new data from the learned model. While the conventional method is stochastic, DDIM (Song et al., 2020a) demonstrates that one can follow a deterministic sampler to generate the final sample $\boldsymbol{x}_0$, which follows the update rule:

$$\boldsymbol{x}_s = (\sigma_s / \sigma_t) \, \boldsymbol{x}_t + (\alpha_s - \alpha_t \sigma_s / \sigma_t) \, \boldsymbol{f}_\phi(\boldsymbol{x}_t, t), \quad s < t, \tag{1}$$

with the boundary condition $\boldsymbol{x}_T = \boldsymbol{\epsilon} \sim \mathcal{N}(0, I)$. As noted in Lu et al. (2022), Eq. (1) is equivalent to the first-order ODE solver for the underlying probability-flow (PF) ODE (Song et al., 2020b). Therefore, the step size $\delta = t - s$ needs to be small to mitigate error accumulation. Additionally, using higher-order solvers such as Runge-Kutta (Süli & Mayers, 2003), Heun (Ascher & Petzold, 1998), and other solvers (Lu et al., 2022; Jolicoeur-Martineau et al., 2021) can further reduce the number of function evaluations (NFEs). However, these approaches are not applicable in single-step.

## 2.2 KNOWLEDGE DISTILLATION

Orthogonal to the development of ODE solvers, distillation-based techniques have been proposed to learn faster student models from a pre-trained diffusion teacher. The most straightforward approach is to perform **direct distillation** (Luhman & Luhman, 2021), where a student model $\boldsymbol{g}_\theta$ is trained to learn from the output of the diffusion model, which is computationally expensive itself:

$$\mathcal{L}_\theta^{\text{Direct}} = \mathbb{E}_{\boldsymbol{\epsilon} \sim \mathcal{N}(0,I)} \| \boldsymbol{g}_\theta(\boldsymbol{\epsilon}) - \texttt{ODE-Solver}(\boldsymbol{f}_\phi, \boldsymbol{\epsilon}, T \to 0) \|_2^2, \tag{2}$$

Here, `ODE-solver` refers to any solvers like DDIM as mentioned above. While this naive approach shows promising results, it typically requires over 50 steps of evaluations to obtain reasonable distillation targets, which becomes a bottleneck when learning large-scale models.

Alternatively, recent studies (Salimans & Ho, 2022; Song et al., 2023; Berthelot et al., 2023) have proposed methods to avoid running the full diffusion path during distillation. For instance, the consistency model (CM, Song et al., 2023) trains a time-conditioned student model $\boldsymbol{g}_\theta(\boldsymbol{x}_t, t)$ to predict self-consistent outputs along the diffusion trajectory in a bootstrap fashion:

$$\mathcal{L}_\theta^{\text{CM}} = \mathbb{E}_{\boldsymbol{x}_t \sim q(\boldsymbol{x}_t | \boldsymbol{x}), s, t \sim [0,T], s < t} \| \boldsymbol{g}_\theta(\boldsymbol{x}_t, t) - \boldsymbol{g}_{\theta^-}(\boldsymbol{x}_s, s) \|_2^2, \tag{3}$$

where $\boldsymbol{x}_s = \texttt{ODE-Solver}(\boldsymbol{f}_\phi, \boldsymbol{x}_t, t \to s)$, typically with a single-step evaluation using Eq. (1). In this case, $\theta^-$ represents an exponential moving average (EMA) of the student parameters $\theta$, which is important to prevent the self-consistency objectives from collapsing into trivial solutions by always predicting similar outputs. After training, samples can be generated by executing $\boldsymbol{g}_\theta(\boldsymbol{x}_T, T)$ with a single NFE. It is worth noting that Eq. (3) requires sampling $\boldsymbol{x}_t$ from the real data sample $\boldsymbol{x}$, which is the essence of bootstrapping: the model learns to denoise increasingly noisy inputs until $\boldsymbol{x}_T$. However, in many tasks, the original training data $\boldsymbol{x}$ for distillation is inaccessible. For example, text-to-image generation models require billions of paired data for training. One possible solution is to use a different dataset for distillation; however, the mismatch in the distributions of the two datasets would result in suboptimal distillation performance.

## 3 METHOD

In this section, we present BOOT, a novel distillation approach inspired by the concept of bootstrapping without requiring target domain data during training. We begin by introducing *signal-ODE*, a modeling technique focused exclusively on signals (§ 3.1), and its corresponding distillation process (§ 3.2). Subsequently, we explore the application of BOOT in text-to-image generation (§ 3.3). The training pipeline is depicted in Fig. 2, providing an overview of the process.

### 3.1 SIGNAL-ODE

We utilize a time-conditioned student model $g_\theta(\epsilon, t)$ in our approach. Similar to direct distillation (Luhman & Luhman, 2021), BOOT always takes random noise $\epsilon$ as input and approximates the intermediate diffusion model variable: $g_\theta(\epsilon, t) \approx x_t = $ `ODE-Solver`$(f_\phi, \epsilon, T \to t), \epsilon \sim \mathcal{N}(0, I)$. This approach eliminates the need to sample from real data during training. The final sample can be obtained as $g_\theta(\epsilon, 0) \approx x_0$. However, it poses a challenge to train $g_\theta$ effectively, as neural networks struggle to predict partially noisy images (Berthelot et al., 2023), leading to out-of-distribution (OOD) problems and additional complexities in learning $g_\theta$ accurately.

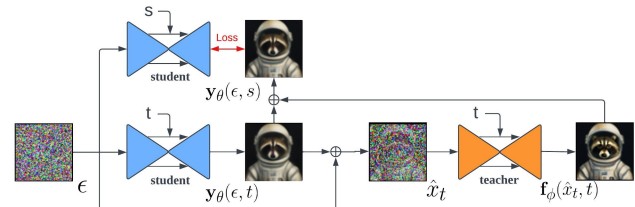

Figure 2: Training pipeline of BOOT. $s$ and $t$ are two consecutive timesteps where $s < t$. From a noise map $\epsilon$, the objective of BOOT minimizes the difference between the output of a student model at timestep $s$, and the output of stacking the same student model and a teacher model at an earlier time $t$. **The whole process is data-free**.

To overcome the aforementioned challenge, we propose an alternative approach where we predict $y_t = (x_t - \sigma_t \epsilon)/\alpha_t$. In this case, $y_t$ represents the low-frequency "signal" component of $x_t$, which is easier for neural networks to learn. The initial noise for diffusion is denoted by $\epsilon$. This prediction target is reasonable since it aligns with the boundary condition of the teacher model, where $y_0 = x_0$. Furthermore, we can derive an iterative equation from Eq. (1) for consecutive timesteps:

$$y_s = \left(1 - e^{\lambda_s - \lambda_t}\right) f_\phi(x_t, t) + e^{\lambda_s - \lambda_t} y_t, \tag{4}$$

where $x_t = \alpha_t y_t + \sigma_t \epsilon$, and $\lambda_t = -\log(\alpha_t/\sigma_t)$ represents the "negative half log-SNR". Notably, the noise term $\epsilon$ automatically cancels out in Eq. (4), indicating that the model always learns from the signal space. Moreover, Eq. (4) demonstrates an interpolation between the current model prediction and the diffusion-denoised output. Similar to the connection between DDIM and PF-ODE (Song et al., 2020b), we can also obtain a continuous version of Eq. (4) by letting $s \to t^-$ as follows:

$$\frac{dy_t}{dt} = -\lambda'_t \cdot (f_\phi(x_t, t) - y_t), \quad y_T \sim p_\epsilon \tag{5}$$

where $\lambda'_t = d\lambda/dt$, and $p_\epsilon$ epresents the boundary distribution of $y_t$. It's important to note that Eq. (5) differs from the PF-ODE, which directly relates to the score function of the data. In our case, the ODE, which we refer to as "Signal-ODE", is specifically defined for signal prediction. At each timestep $t$, a fixed noise $\epsilon$ is injected and denoised by the diffusion model $f_\phi$. The Signal-ODE implies a "ground-truth" trajectory for sampling new data. For example, one can initialize a reasonable $y_T = \epsilon \sim \mathcal{N}(0, I)$ and solve the Signal-ODE to obtain the final output $y_0$. Although the computational complexity remains the same as conventional DDIM, we will demonstrate in the next section how we can efficiently approximate $y_t$ using bootstrapping objectives.

## 3.2 LEARNING WITH BOOTSTRAPPING

Our objective is to learn $y_\theta(\epsilon, t) \approx y_t$ as a single-step prediction model using neural networks, rather than solving the signal-ODE with Eq. (5). By matching both sides of Eq. (5), we obtain the objective:

$$\mathcal{L}_\theta^{DE} = \mathbb{E}_{\epsilon \sim \mathcal{N}(0, I), t \sim [0, T]} \left\| \frac{dy_\theta(\epsilon, t)}{dt} + \lambda'_t \cdot (f_\phi(\hat{x}_t, t) - y_\theta(\epsilon, t)) \right\|_2^2. \tag{6}$$

In Eq. (6), we use $y_\theta(\epsilon, t)$ to estimate $y_t$, and $\hat{x}_t = \alpha_t y_\theta(\epsilon, t) + \sigma_t \epsilon$ represents the corresponding noisy image. Instead of using forward-mode auto-differentiation, which can be computationally expensive, we can approximate the above equation with finite differences due to the 1-dimensional nature of $t$. The approximate form is similar to Eq. (4):

$$\mathcal{L}_\theta^{BS} = \mathbb{E}_{\epsilon \sim \mathcal{N}_{0,I}, t \sim [\delta, T]} \left[ \frac{\tilde{w}_t}{\delta^2} \left\| y_\theta(\epsilon, s) - \text{SG}\left[ y_\theta(\epsilon, t) + \underbrace{\delta \lambda'_t \cdot ((f_\phi(\hat{x}_t, t)) - y_\theta(\epsilon, t))}_{\text{incremental improvement}} \right] \right\|_2^2 \right], \tag{7}$$

where $s = t - \delta$ and $\delta$ is the discrete step size. $\tilde{w}_t$ represents the time-dependent loss weighting, which can be chosen uniformly. We use SG[.] as the stop-gradient operator for training stability.

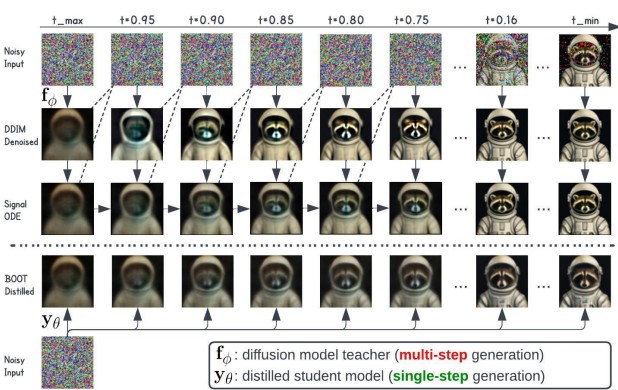

Unlike CM-based methods, such as those mentioned in Eq. (3), we do not require an exponential moving average (EMA) copy of the student parameters to avoid collapsing. This avoids potential slow convergence and sub-optimal solutions. As shown in Eq. (7), the proposed objective is unlikely to degenerate because there is an incremental improvement term in the training target, which is mostly non-zero. In other words, we can consider $\boldsymbol{y}_\theta$ as an exponential moving average of $\boldsymbol{f}_\phi$, with a decaying rate of $1 - \delta\lambda'_t$. This ensures that the student model always receives distinguishable signals for different values of $t$.

Figure 3: Comparison between the generated outputs of DDIM/Signal-ODE and our distilled model given the same prompt. By definition, signal-ODE converges to the same final sample as the original DDIM, while the distilled single-step model does not necessarily follow.

**Error Accumulation**  One critical challenge in learning BOOT is the "error accumulation" issue, where imperfect predictions of $\boldsymbol{y}_\theta$ on large $t$ can propagate to subsequent timesteps. While similar challenges exist in other bootstrapping-based approaches, it becomes more pronounced in our case due to the possibility of out-of-distribution inputs $\hat{\boldsymbol{x}}_t$ for the teacher model, resulting from error accumulation and leading to incorrect learning signals. To mitigate this, we employ two methods: (1) We uniformly sample $t$ throughout the training time, despite the potential slowdown in convergence. (2) We use a higher-order solver (e.g., Heun's method (Ascher & Petzold, 1998)) to compute the bootstrapping target with better estimation.

**Boundary Condition**  In theory, the boundary $\boldsymbol{y}_T$ can have arbitrary values since $\alpha_T = 0$, and the value of $\boldsymbol{y}_T$ does not affect the value $\boldsymbol{x}_T = \boldsymbol{\epsilon}$. However, $\lambda'_t$ is unbounded at $t = T$, leading to numerical issues in optimization. As a result, the student model must be learned within a truncated range $t \in [t_{\min}, t_{\max}]$. This necessitates additional constraints at the boundaries to ensure that $\alpha_{t_{\max}}\boldsymbol{y}_\theta(\boldsymbol{\epsilon}, t_{\max}) + \sigma_{t_{\max}}\boldsymbol{\epsilon}$ follows the same distribution as the diffusion model. In this work, we address this through an auxiliary boundary loss:

$$\mathcal{L}_\theta^{\text{BC}} = \mathbb{E}_{\boldsymbol{\epsilon} \sim \mathcal{N}(0, I)} \left[ \|\boldsymbol{f}_\phi(\boldsymbol{\epsilon}, t_{\max}) - \boldsymbol{y}_\theta(\boldsymbol{\epsilon}, t_{\max})\|_2^2 \right]. \tag{8}$$

Here, we enforce the student model to match the initial denoising output. In our early exploration, we found that the boundary condition is crucial for the single-step student to fully capture the modeling space of the teacher, especially in text-to-image scenarios. Failure to learn the boundaries tends to result in severe mode collapse and color-saturation problems.

The overall learning objective combines $\mathcal{L}_\theta = \mathcal{L}_\theta^{\text{BS}} + \beta\mathcal{L}_\theta^{\text{BC}}$, where $\beta$ is a hyper-parameter. The algorithm for student model distillation is presented in Appendix Algorithm 1.

## 3.3 DISTILLATION OF TEXT-TO-IMAGE MODELS

Our approach can be readily applied for distilling conditional diffusion models, such as text-to-image generation (Ramesh et al., 2022; Rombach et al., 2021; Balaji et al., 2022), where a conditional denoiser $\boldsymbol{f}_\phi(\boldsymbol{x}_t, t, \boldsymbol{c})$ is learned with the same objective given an aligned dataset. In practice, inference of these models requires necessary post-processing steps for amplifying the conditional generation. For instance, classifier-free guidance (CFG, Ho & Salimans, 2022) can be applied as:

$$\tilde{\boldsymbol{f}}_\phi(\boldsymbol{x}_t, t, \boldsymbol{c}) = \boldsymbol{f}_\phi(\boldsymbol{x}_t, t, \boldsymbol{n}) + w \cdot (\boldsymbol{f}_\phi(\boldsymbol{x}_t, t, \boldsymbol{c}) - \boldsymbol{f}_\phi(\boldsymbol{x}_t, t, \boldsymbol{n})), \tag{9}$$

where $\boldsymbol{n}$ is the negative prompt (or empty), and $w$ is the guidance weight (by default $w = 7.5$) over the denoised signals. We directly use the modified $\tilde{\boldsymbol{f}}_\phi$ to replace the original $\boldsymbol{f}_\phi$ in the training objectives in Eqs. (7) and (8). Optionally, similar to Meng et al. (2022), we can also learn student model condition on both $t$ and $w$ to reflect different guidance strength.

Our method can be easily adopted in either pixel (Saharia et al., 2022) or latent space (Rombach et al., 2021) models without any change in implementation. For pixel-space models, it is sometimes critical

|  | Steps | FFHQ $64 \times 64$ FID / Prec. / Rec. | fps | LSUN $256 \times 256$ FID / Prec. / Rec. | fps | ImageNet $64 \times 64$ FID / Prec. / Rec. | fps |
|---|---|---|---|---|---|---|---|
| DDPM | 250 | 5.4 / 0.80 / 0.54 | 0.2 | 8.2 / 0.55 / 0.43 | 0.1 | 11.0 / 0.67 / 0.58 | 0.1 |
| DDIM | 50 | 7.6 / 0.79 / 0.48 | 1.2 | 13.5 / 0.47 / 0.40 | 0.6 | 13.7 / 0.65 / 0.56 | 0.6 |
|  | 10 | 18.3 / 0.78 / 0.27 | 5.3 | 31.0 / 0.27 / 0.32 | 3.1 | 18.3 / 0.60 / 0.49 | 3.3 |
|  | 1 | 225 / 0.10 / 0.00 | 54 | 308 / 0.00 / 0.00 | 31 | 237 / 0.05 / 0.00 | 34 |
| Ours | 1 | 9.0 / 0.79 / 0.38 | 54 | 23.4 / 0.38 / 0.29 | 32 | 12.3 / 0.69 / 0.46 | 34 |

Table 1: Comparison for image generation benchmarks on FFHQ, LSUN and class-conditioned ImageNet. For ImageNet, numbers are reported without using CFG ($w = 1$).

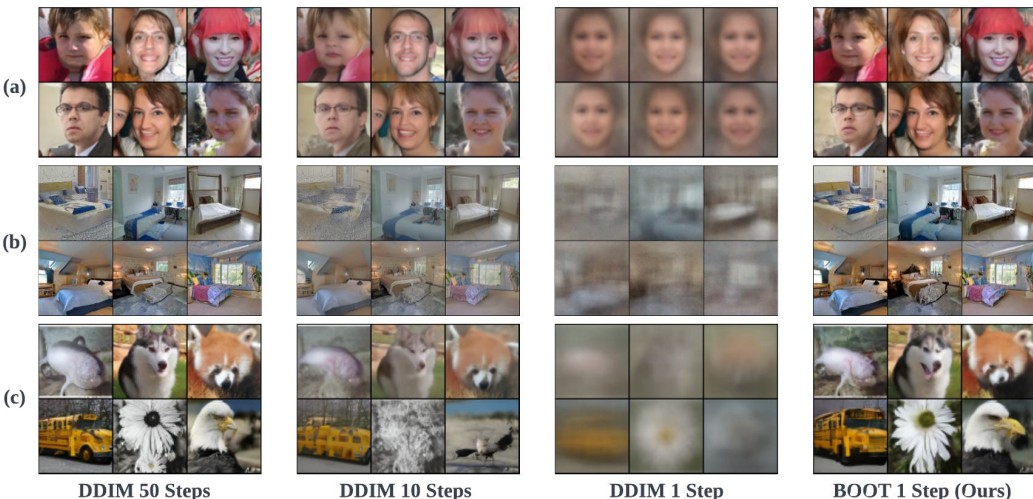

**DDIM 50 Steps**  **DDIM 10 Steps**  **DDIM 1 Step**  **BOOT 1 Step (Ours)**

Figure 4: Uncurated samples of {50, 10, 1} DDIM sampling steps and the proposed BOOT from (a) FFHQ (b) LSUN (c) ImageNet benchmarks, respectively, given the same set of initial noise input.

to apply clipping or dynamic thresholding (Saharia et al., 2022) over the denoised targets to avoid over-saturation. Similarly, we also clip the targets in our objectives Eqs. (7) and (8). Pixel-space models (Saharia et al., 2022) typically involve learning cascaded models (one base model + a few super-resolution (SR) models) to increase the output resolutions progressively. We can also distill the SR models with BOOT into one step by conditioning both the SR teacher and the student with the output of the distilled base model.

## 4 EXPERIMENTS

### 4.1 EXPERIMENTAL SETUPS

**Diffusion Model Teachers** We begin by evaluating the performance of BOOT on diffusion models trained on standard image generation benchmarks: CIFAR-10 $32 \times 32$ (Krizhevsky et al., 2009), FFHQ $64 \times 64$ (Karras et al., 2017), class-conditional ImageNet $64 \times 64$ (Deng et al., 2009), LSUN Bedroom $256 \times 256$ (Yu et al., 2015). On CIFAR-10, we compare with other established methods, including PD (Salimans & Ho, 2022), CM (Song et al., 2023) trained with 800K iterations, as well as fast sampling solvers. For these experiments, we adopt the EDM teacher (Karras et al., 2022). For other datasets, we train the teacher diffusion models separately using signal prediction objectives.

For text-to-image generation scenarios, we directly apply BOOT to open-sourced diffusion models in both pixel-space (DeepFloyd-IF (IF), Saharia et al., 2022) and latents space (StableDiffusion (SD), Rombach et al., 2021). Thanks to the data-free nature of BOOT, we do not require access to the original training set, which may consist of billions of text-image pairs with unknown preprocessing steps. Instead, we only need the prompt conditions to distill both models. In this work, we consider general-purpose prompts generated by users. Specifically, we utilize diffusiondb (Wang et al., 2022),

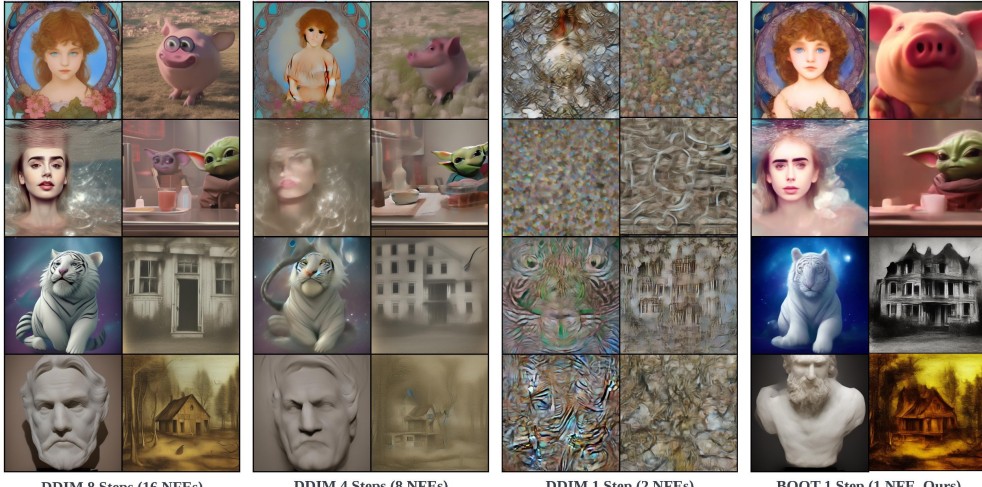

Figure 5: Uncurated samples of {8, 4, 1} DDIM sampling steps and the proposed BOOT from SD2.1-base, given the same set of initial noise input and prompts sampled from *diffusiondb*.

a large-scale prompt dataset that contains 14 million images generated by StableDiffusion using prompts provided by real users. We only use the text prompts for distillation.

**Implementation Details** Similar to previous research (Song et al., 2023), we use student models with architectures similar to those of the teachers, having nearly identical numbers of parameters. A more comprehensive architecture search is left for future work. We initialize the majority of the student $y_\theta$ parameters with the teacher model $f_\phi$, except for the newly introduced conditioning modules (target timestep $t$ and potentially the CFG weight $w$), which are incorporated into the U-Net architecture in a similar manner as how class labels were incorporated. It is important to note that the target timestep $t$ is different from the original timestep used for conditioning the diffusion model, which is always set to $t_{max}$ for the student model. Additionally, for CIFAR-10 experiments, we also train our models coupled with LPIPS loss (Zhang et al., 2018).

**Evaluation Metrics** For image generation, results are compared according to Fréchet Inception Distance (FID, Heusel et al., 2017), Precision and Recall (Kynkäänniemi et al., 2019)

| Method | NFE | FID |
|---|---|---|
| **Diffusion Model+Solver** | | |
| DDPM (Ho et al., 2020) | 1000 | 3.17 |
| DDIM (Song et al., 2021) | 50 | 4.67 |
| DPM-solver-2 (Lu et al., 2022) | 10 | 5.94 |
| DEIS (Zhang & Chen, 2022) | 10 | 4.17 |
| EDM (Karras et al., 2022) | 35 | 2.04 |
| **Distillation** | | |
| Direct* (Luhman & Luhman, 2021) | 1 | 9.36 |
| DFNO* (Zheng et al., 2023) | 1 | 4.12 |
| ReFlow* (Liu et al., 2022) | 1 | 6.18 |
| PD (Salimans & Ho, 2022) | 1 | 8.34 |
| CM (Song et al., 2023) | 1 | 3.55 |
| **Data-free Distillation** | | |
| Ours (L2 loss) | 1 | 6.88 |
| Ours (LPIPS loss) | 1 | 4.38 |

Table 2: Unconditional image generation on the CIFAR-10 dataset. * indicates methods requiring synthesizing additional dataset.

over 50, 000 real samples from the corresponding datasets. For text-to-image tasks, we measure the zero-shot CLIP score (Radford et al., 2021) for measuring the faithfulness of generation given 5000 randomly sampled captions from COCO2017 (Lin et al., 2014) validation set. In addition, we also report the inference speed measured by fps with batch-size 1 on a single A100 GPU.

## 4.2 RESULTS

**Quantitative Results** We first evaluate the proposed method on standard image generation benchmarks. The quantitative comparison with the standard diffusion inference methods like DDPM (Ho et al., 2020) and the deterministic DDIM (Song et al., 2020a) are shown in Table 1. Despite lagging behind the 50-step DDIM inference, BOOT significantly improves the performance 1-step inference,

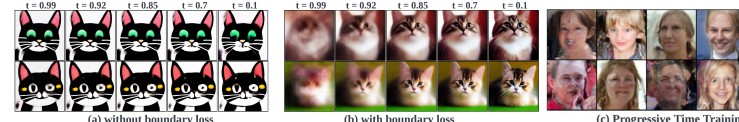

Figure 6: Ablation Study. (a) vs. (b): The additional boundary loss in § 3.2 alleviates the mode collapsing issue and prompts diversity in generation. (c) vs. (d): Uniform time training yields better generation compared with progressive time training.

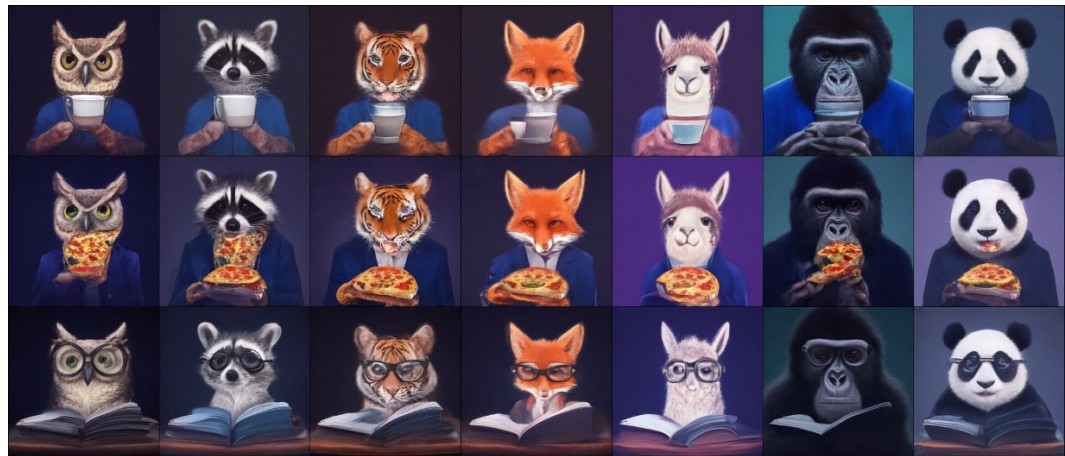

Figure 7: With fixed noise, we can perform controllable generation by swapping the keywords from the prompts. The prompts are chosen from the combination of *portrait of a {owl, raccoon, tiger, fox, llama, gorilla, panda} wearing {a t-shirt, a jacket, glasses} {drinking a latte, eating a pizza, reading a book} cinematic, hdr.* All images are generated from the student distilled from IF teacher.

and achieves better performance against DDIM with around 10 denoising steps, while maintaining $\times 10$ speed-up. Note that, the speed advantage doubles if the teacher employs guidance.

Also, we show quantitative results on CIFAR-10 compared against existing methods in Table 2. It is important to highlight that none of the considered distillation approaches can be categorized as completely data-free. These methods either necessitate the generation of expansive synthetic datasets or depend on real data sources. Our approach surpasses PD and boasts comparable results when contrasted with CM (which was trained much longer than our models). Note that our approach is the first to achieve data-free training to enable highly efficient single-step generation.

Additionally, we conduct quantitative evaluation on text-to-image tasks. Using the SD teacher, we obtain a CLIP-score of $0.254$ on COCO2017, a slight degradation compared to the 50-step DDIM results ($0.262$), while it generates 2 orders of magnitude faster, rendering real-time applications.

**Visual Results** We show the qualitative comparison in Figs. 4 and 5 for image generation and text-to-image, respectively. For both cases, naïve 1-step inference fails completely, and the diffusion generally outputs almost empty and ill-structured images with fewer than 10 NFEs. In contrast, BOOT is able to synthesize high-quality images that are visually close (Fig. 4) or semantically similar (Fig. 5) to teacher's results with much more steps. Unlike the standard benchmarks, distilling text-to-image models (e.g., SD) typically leads to noticeably different generation from the original diffusion model, even starting with the same initial noise. We hypothesize it is a combined effect of highly complex underlying distribution and CFG. We show more results including pixel-space models in the appendix.

### 4.3 ANALYSIS

**Importance of Boundary Condition** The significance of incorporating the boundary loss is demonstrated in Fig. 6 (a) and (b). When using the same noise inputs, we compare the student outputs based on different target timesteps. As $\boldsymbol{y}_\theta(\boldsymbol{\epsilon}, t)$ tracks the signal-ODE output, it produces more averaged

results as $t$ approaches 1. However, without proper boundary constraints, the student outputs exhibit consistent sharpness across timesteps, resulting in over-saturated and non-realistic images. This indicates a complete failure of the learned student model to capture the distribution of the teacher model, leading to severe mode collapse.

**Progressive v.s. Uniform Time Training** We also compare different training strategies in Fig. 6 (c) and (d). In contrast to the proposed approach of uniformly sampling $t$, one can potentially achieve additional efficiency with a fixed schedule that progressively decreases $t$ as training proceeds. This progressive training strategy seems reasonable considering that the student is always initialized from $t_{\max}$ and gradually learns to predict the clean signals (small $t$) during training. However, progressive training tends to introduce more artifacts (as observed in the visual comparison in Fig. 6). We hypothesize that progressive training is more prone to accumulating irreversible errors.

**Controllable Generation** In Fig. 7, we provide an example of text-controlled generation by fixing the noise input and only modifying the prompts. Similar to the original diffusion teacher model, the BOOT distilled student retains the ability of disentangled representation, enabling fine-grained control while maintaining consistent styles. Additionally, in Appendix Fig. 13, we visualize the results of latent space interpolation, where the student model is distilled from the pretrained IF teacher. The smooth transition of the generated images demonstrates that the distilled student model has successfully learned a continuous and meaningful latent space.

## 5 RELATED WORK

**Improving Efficiency of Diffusion Models** Speeding up inference of diffusion models is a broad area. Recent works and also our work (Luhman & Luhman, 2021; Salimans & Ho, 2022; Meng et al., 2022; Song et al., 2023; Berthelot et al., 2023) aim at reducing the number of diffusion model inference steps via distillation. Aside from distillation methods, other representative approaches include advanced ODE solvers (Karras et al., 2022; Lu et al., 2022), low-dimension space diffusion (Rombach et al., 2021; Vahdat et al., 2021; Jing et al., 2022; Gu et al., 2022), and improved diffusion targets (Lipman et al., 2023; Liu et al., 2022). BOOT is orthogonal and complementary to these approaches, and can theoretically benefit from improvements made in all these aspects.

**Knowledge Distillation for Generative Models** Knowledge distillation (Hinton et al., 2015) has seen successful applications in learning efficient generative models, including model compression (Kim & Rush, 2016; Aguinaldo et al., 2019; Fu et al., 2020; Hsieh et al., 2023) and non-autoregressive sequence generation (Gu et al., 2017; Oord et al., 2018; Zhou et al., 2019). We believe that BOOT could inspire a new paradigm of distilling powerful generative models without requiring access to the training data.

## 6 DISCUSSION AND CONCLUSION

**Limitations** BOOT may produce lower quality samples compared to other distillation methods (Song et al., 2023; Berthelot et al., 2023) which require ground-truth data for training. This issue can potentially be remedied by combining BOOT with these methods. Another limitation is that the current design only focuses on data-free distillation into a single-step student model and cannot support multi-step generation as did in previous work (Song et al., 2023) for further quality improvement.

As future research, we aim to investigate the possibility of jointly training the teacher and the student models in a manner that incorporates the concept of diffusion into the distillation process. Furthermore, we find it intriguing to explore the training of a single-step diffusion model from scratch. This exploration could provide insights into the applicability and benefits of BOOT in scenarios where a pre-trained model is not available. Finally, extending BOOT to multi-step generation is also feasible, which can be achieved by training the student with multiple timesteps coupled with restart sampling (Xu et al., 2023) approaches.

**Conclusion** In summary, this paper introduced a novel technique *BOOT* to distill diffusion models into single-step models. The method did not require the presence of any real or synthetic data by learning a time-conditioned student model with bootstrapping objectives. The proposed approach achieved comparable generation quality while being significantly faster, compared to the diffusion teacher, and was also applicable to large-scale text-to-image generation, showcasing its versatility.

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

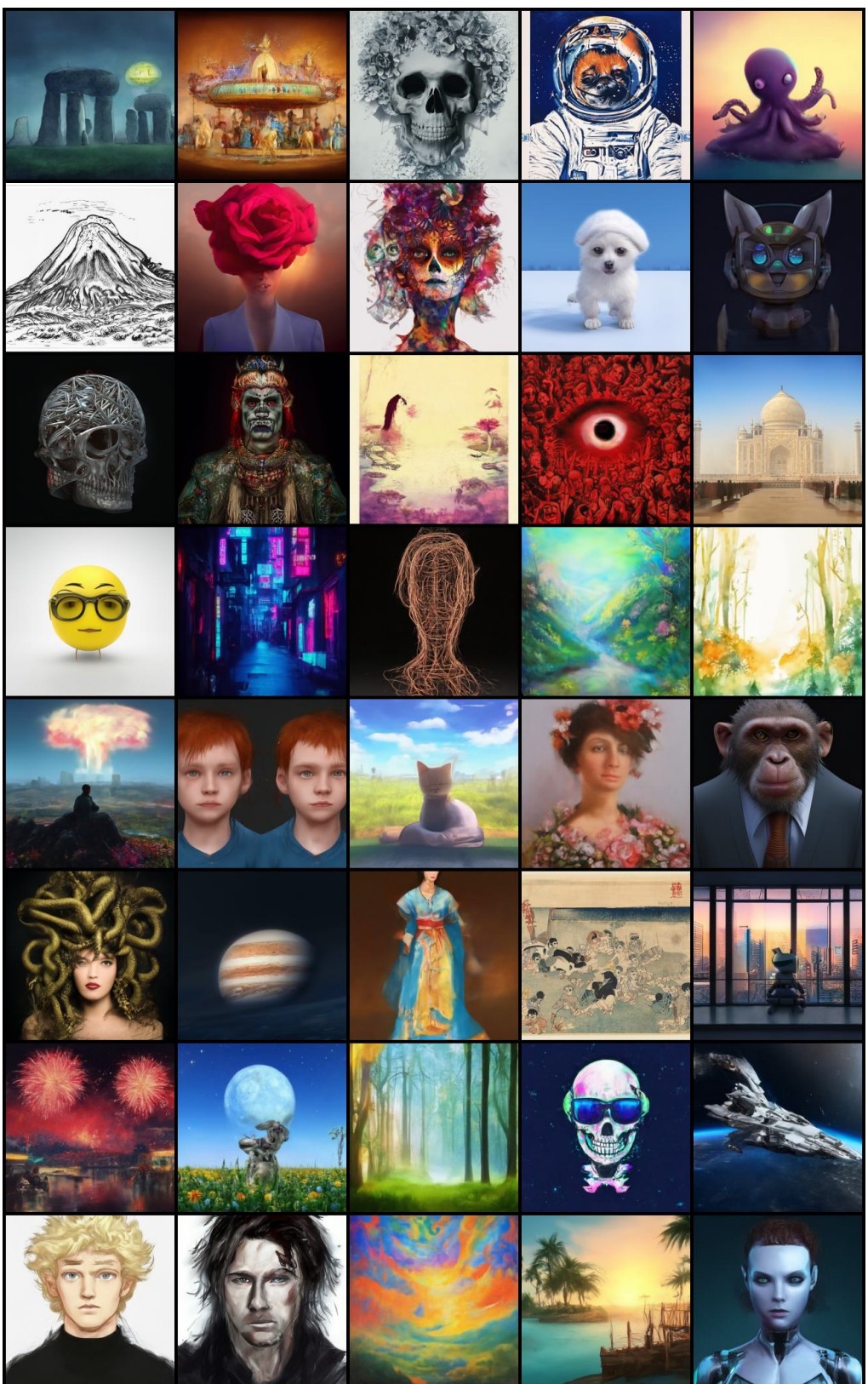

Figure 8: Samples of our distilled single-step model with prompts from *diffusiondb*.

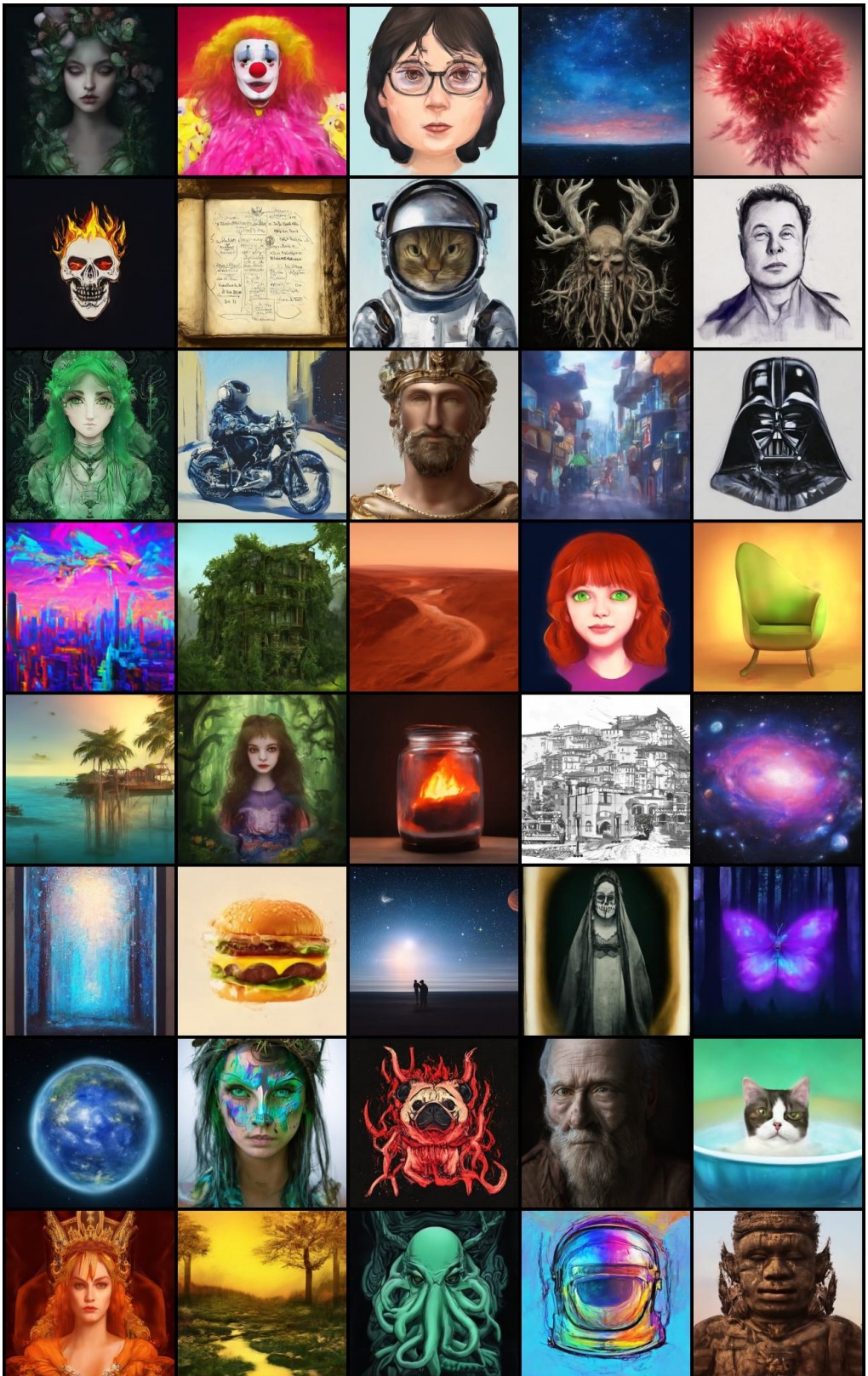

Figure 9: Samples of our distilled single-step model with prompts from *diffusiondb*.

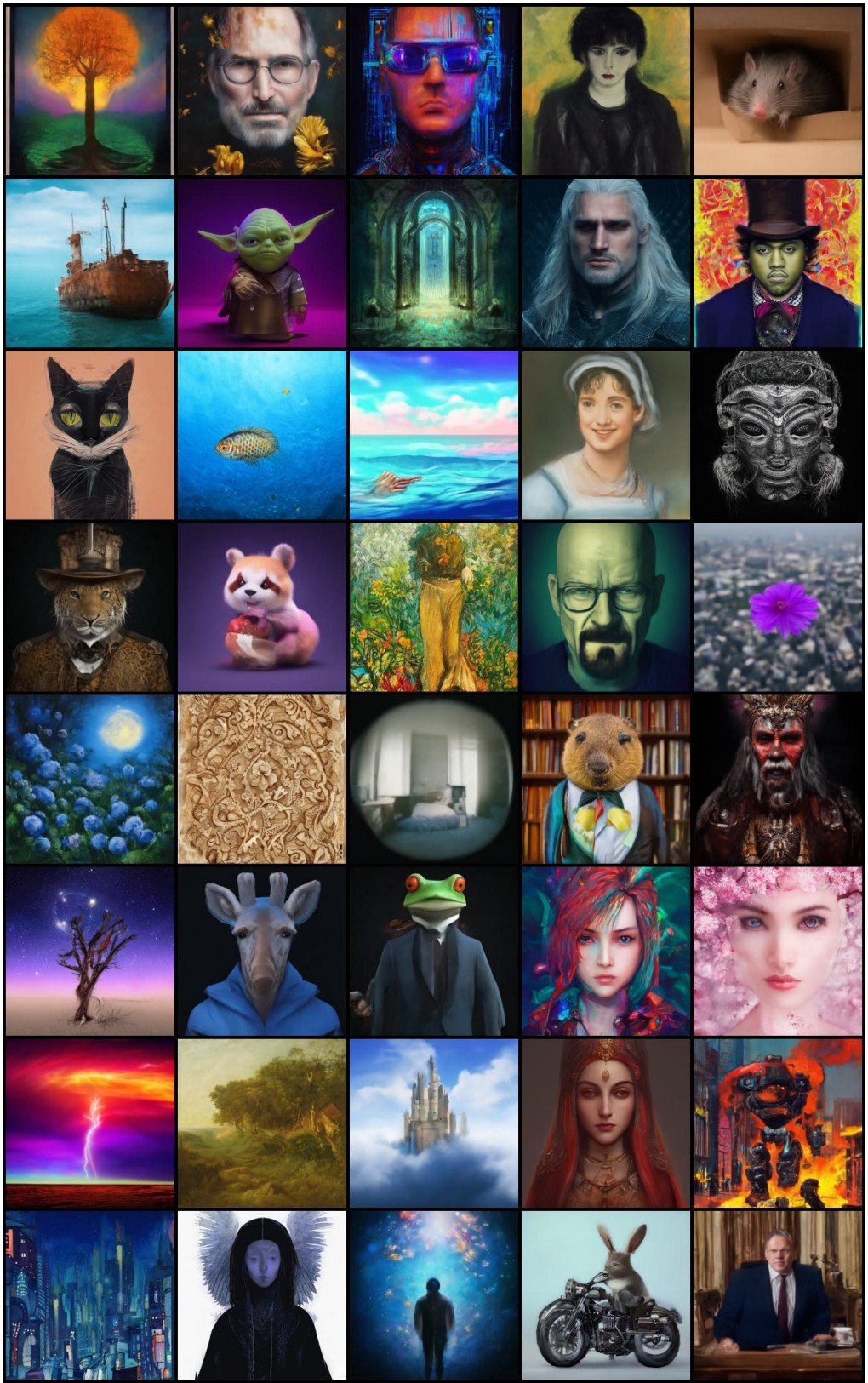

Figure 10: Samples of our distilled single-step model with prompts from *diffusiondb*.

# Contents

# Appendices

## A ALGORITHM DETAILS

### A.1 NOTATIONS

In this paper, we use $\boldsymbol{f}_\phi(\boldsymbol{x}, t)$ to represent the diffusion model that denoises the noisy sample $\boldsymbol{x}$ into its clean version, and we derive the DDIM sampler (Eq. (1)) following the definition of Song et al. (2020a): we deterministically synthesize $\boldsymbol{x}_s$ based on the following update rule:

$$
\begin{aligned}
\boldsymbol{x}_s &= \texttt{ODE-Solver}(\boldsymbol{f}_\phi, \boldsymbol{\epsilon}, T \to s) \\
&= \alpha_s \boldsymbol{f}_\phi(\boldsymbol{x}_t, t) + \sigma_s \left( \frac{\boldsymbol{x}_t - \alpha_t \boldsymbol{f}_\phi(\boldsymbol{x}_t, t)}{\sigma_t} \right) \\
&= \frac{\sigma_s}{\sigma_t} \boldsymbol{x}_t + \left( \alpha_s - \alpha_t \frac{\sigma_s}{\sigma_t} \right) \boldsymbol{f}_\phi(\boldsymbol{x}_t, t)
\end{aligned}
\tag{10}
$$

where $0 \le s < t \le T$. Here we use $\texttt{ODE-Solver}$ to represent the DDIM sampling from a random noise $\boldsymbol{x}_T = \boldsymbol{\epsilon} \sim \mathcal{N}(0, I)$, and iteratively obtain the sample at step $s$. In practice, we can generalize to higher-order ODE-solvers for better efficiency.

For distillation, we define the student model with $\boldsymbol{g}_\theta(\boldsymbol{\epsilon}, t)$ which approximates $\boldsymbol{x}_t$ along the diffusion trajectory above. To avoid directly predicting the noisy samples $\boldsymbol{x}_t$ with neural networks, we re-parameterize $\boldsymbol{g}_\theta(\boldsymbol{\epsilon}, t) = \alpha_t \boldsymbol{y}_\theta(\boldsymbol{\epsilon}, t) + \sigma_t \boldsymbol{\epsilon}$ where the noise part is constant throughout $t$ except the scale factor $\sigma_t$. In this way, the learning goal $\boldsymbol{y}_\theta(\boldsymbol{\epsilon}, t)$ is to predict a new variable $\boldsymbol{y}_t$: the "signal" part of the original variable $\boldsymbol{y}_t = (\boldsymbol{x}_t - \sigma_t \boldsymbol{\epsilon})/\alpha_t$.

### A.2 DERIVATION OF SIGNAL-ODE

Based on the definition of $\boldsymbol{y}_t = (\boldsymbol{x}_t - \sigma_t \boldsymbol{\epsilon})/\alpha_t$, we can derive the following equations from Eq. (10):

$$
\begin{aligned}
\boldsymbol{x}_s &= \frac{\sigma_s}{\sigma_t} \boldsymbol{x}_t + \left( \alpha_s - \alpha_t \frac{\sigma_s}{\sigma_t} \right) \boldsymbol{f}_\phi(\boldsymbol{x}_t, t) \\
\Rightarrow \alpha_s \boldsymbol{y}_s + \sigma_s \boldsymbol{\epsilon} &= \frac{\sigma_s}{\sigma_t} (\alpha_t \boldsymbol{y}_t + \sigma_t \boldsymbol{\epsilon}) + \left( \alpha_s - \alpha_t \frac{\sigma_s}{\sigma_t} \right) \boldsymbol{f}_\phi(\boldsymbol{x}_t, t) \\
\Rightarrow \alpha_s \boldsymbol{y}_s + \cancel{\sigma_s \boldsymbol{\epsilon}} &= \alpha_t \frac{\sigma_s}{\sigma_t} \boldsymbol{y}_t + \cancel{\sigma_s \boldsymbol{\epsilon}} + \left( \alpha_s - \alpha_t \frac{\sigma_s}{\sigma_t} \right) \boldsymbol{f}_\phi(\boldsymbol{x}_t, t) \\
\Rightarrow \boldsymbol{y}_s &= \frac{\alpha_t \sigma_s}{\sigma_t \alpha_s} \boldsymbol{y}_t + \left( 1 - \frac{\alpha_t \sigma_s}{\sigma_t \alpha_s} \right) \boldsymbol{f}_\phi(\boldsymbol{x}_t, t) \\
&= \left( 1 - e^{\lambda_s - \lambda_t} \right) \boldsymbol{f}_\phi(\boldsymbol{x}_t, t) + e^{\lambda_s - \lambda_t} \boldsymbol{y}_t,
\end{aligned}
\tag{11}
$$

where we use the auxiliary variable $\lambda_t = -\log(\alpha_t/\sigma_t)$ for simplifying the equations. As mentioned in § 3.1, we can further obtain the continuous form of Eq. (11) by assigning $t - s \to 0$. That is, Eq. (11) is equivalent to that shown in the following:

$$
\begin{aligned}
\boldsymbol{y}_s &= \left( 1 - e^{\lambda_s - \lambda_t} \right) \boldsymbol{f}_\phi(\boldsymbol{x}_t, t) + e^{\lambda_s - \lambda_t} \boldsymbol{y}_t \\
\Rightarrow \boldsymbol{y}_t - \boldsymbol{y}_s &= - \left( 1 - e^{\lambda_s - \lambda_t} \right) (\boldsymbol{f}_\phi(\boldsymbol{x}_t, t) - \boldsymbol{y}_t) \\
\Rightarrow \frac{\boldsymbol{y}_t - \boldsymbol{y}_s}{t - s} &= - \frac{e^{\lambda_t} - e^{\lambda_s}}{t - s} \cdot e^{-\lambda_t} (\boldsymbol{f}_\phi(\boldsymbol{x}_t, t) - \boldsymbol{y}_t) \\
\Rightarrow \frac{\mathrm{d}\boldsymbol{y}_t}{\mathrm{d}t} &= - \cancel{e^{\lambda_t}} \cdot \lambda_t' \cdot \cancel{e^{-\lambda_t}} (\boldsymbol{f}_\phi(\boldsymbol{x}_t, t) - \boldsymbol{y}_t)
\end{aligned}
\tag{12}
$$

where $\lambda_t' = \mathrm{d}\lambda_t/\mathrm{d}t$. Given a fixed noise input $\boldsymbol{\epsilon}$, Eq. (12) defines an ODE over $\boldsymbol{y}_\theta$ w.r.t $t$, which we call *Signal-ODE*, as both sides of the equation only operate in "low-frequency" signal space.

---

**Algorithm 1** Distillation using BOOT for Conditional Diffusion Models.

---

**Require:** pretrained diffusion model $\boldsymbol{f}_\phi$, initial student parameter from the teacher $\theta \leftarrow \phi$, step size $\delta$, learning rate $\eta$, CFG weight $w$, context dataset $\mathcal{D}$, negative condition $\boldsymbol{n} = \emptyset$, $t_{\min}, t_{\max}, \beta$.

1: **while** not converged **do**
2:      Sample noise input $\boldsymbol{\epsilon} \sim \mathcal{N}(0, I)$
3:      Sample context input $\boldsymbol{c} \sim \mathcal{D}$
4:      Sample $t \sim (t_{\min}, t_{\max}), s = \min(t - \delta, t_{\min})$
5:      Compute noise schedule $\alpha_t, \sigma_t, \alpha_s, \sigma_s$
6:      Compute $\lambda'_t \approx (1 - \frac{\alpha_t \sigma_s}{\sigma_t \alpha_s})/\delta$
7:      Generate the model predictions:
8:          $\boldsymbol{y}_t = \boldsymbol{y}_\theta(\boldsymbol{\epsilon}, t, \boldsymbol{c}), \quad \boldsymbol{y}_s = \boldsymbol{y}_\theta(\boldsymbol{\epsilon}, s, \boldsymbol{c}), \quad \boldsymbol{y}_{t_{\max}} = \boldsymbol{y}_\theta(\boldsymbol{\epsilon}, t_{\max}, \boldsymbol{c})$
9:      Generate the noisy sample $\hat{\boldsymbol{x}}_t = \alpha_t \boldsymbol{y}_t + \sigma_t \boldsymbol{\epsilon}$
10:     Compute the denoised target:
11:         $\tilde{\boldsymbol{f}}_t = \boldsymbol{f}_\phi(\hat{\boldsymbol{x}}_t, t, \boldsymbol{n}) + w \cdot (\boldsymbol{f}_\phi(\hat{\boldsymbol{x}}_t, t, \boldsymbol{c}) - \boldsymbol{f}_\phi(\hat{\boldsymbol{x}}_t, t, \boldsymbol{n}))$
12:         $\tilde{\boldsymbol{f}}_{t_{\max}} = \boldsymbol{f}_\phi(\boldsymbol{\epsilon}, t_{\max}, \boldsymbol{n}) + w \cdot (\boldsymbol{f}_\phi(\boldsymbol{\epsilon}, t_{\max}, \boldsymbol{c}) - \boldsymbol{f}_\phi(\boldsymbol{\epsilon}, t_{\max}, \boldsymbol{n}))$
13:     Compute the bootstrapping loss $\mathcal{L}_\theta^{\text{BS}} = \frac{1}{(\delta\lambda'_t)^2} \|\boldsymbol{y}_s - \text{SG}(\boldsymbol{y}_t + \delta\lambda'_t(\tilde{\boldsymbol{f}}_t - \boldsymbol{y}_t))\|_2^2$
14:     Compute the boundary loss $\mathcal{L}_\theta^{\text{BC}} = \|\boldsymbol{y}_{t_{\max}} - \tilde{\boldsymbol{f}}_{t_{\max}}\|_2^2$
15:     Update model parameters $\theta \leftarrow \theta - \eta \cdot \nabla_\theta \left( \mathcal{L}_\theta^{\text{BS}} + \beta\mathcal{L}_\theta^{\text{BC}} \right)$
16: **end while**
17: **return** Trained model parameters $\theta$

---

### A.3    BOOTSTRAPPING OBJECTIVES

The bootstrapping objectives in Eq. (7) can be easily derived by taking the finite difference of Eq. (3). Here we use $\boldsymbol{y}_\theta(\boldsymbol{\epsilon}, t)$ to estimate $\boldsymbol{y}_t$, and use $\hat{\boldsymbol{x}}_t$ to represent the noisy image obtained from $\boldsymbol{y}_\theta(\boldsymbol{\epsilon}, t)$.

$$
\begin{aligned}
\mathcal{L}_\theta &= \mathbb{E}_{\boldsymbol{\epsilon},t} \left[ \tilde{\omega}_t \left\| \frac{\mathrm{d}\boldsymbol{y}_\theta(\boldsymbol{\epsilon}, t)}{\mathrm{d}t} + \lambda'_t \cdot (\boldsymbol{f}_\phi(\hat{\boldsymbol{x}}_t, t) - \boldsymbol{y}_\theta(\boldsymbol{\epsilon}, t)) \right\|_2^2 \right] \\
&\approx \mathbb{E}_{\boldsymbol{\epsilon},t} \left[ \tilde{\omega}_t \| \frac{\boldsymbol{y}_\theta(\boldsymbol{\epsilon}, s) - \boldsymbol{y}_\theta(\boldsymbol{\epsilon}, t)}{\delta} - \lambda'_t (\boldsymbol{f}_\phi(\hat{\boldsymbol{x}}_t, t) - \boldsymbol{y}_\theta(\boldsymbol{\epsilon}, t)) \|_2^2 \right] \\
&= \mathbb{E}_{\boldsymbol{\epsilon},t} \left[ \frac{\tilde{\omega}_t}{\delta^2} \| \boldsymbol{y}_\theta(\boldsymbol{\epsilon}, s) - [\boldsymbol{y}_\theta(\boldsymbol{\epsilon}, t) + \delta\lambda'_t (\boldsymbol{f}_\phi(\hat{\boldsymbol{x}}_t, t) - \boldsymbol{y}_\theta(\boldsymbol{\epsilon}, t))] \|_2^2 \right] \\
&= \mathbb{E}_{\boldsymbol{\epsilon},t} \left[ \frac{\tilde{\omega}_t}{\delta^2} \| \boldsymbol{y}_\theta(\boldsymbol{\epsilon}, s) - \hat{\boldsymbol{y}}_\theta(\boldsymbol{\epsilon}, s) \|_2^2 \right],
\end{aligned}
\tag{13}
$$

where $s = t - \delta$, and $\hat{\boldsymbol{y}}_\theta(\boldsymbol{\epsilon}, s)$ is the approximated target. $\tilde{\omega}_t$ is the additional weight, where by default $\tilde{\omega}_t = 1$. To stabilize training, a stop-gradient operation $\text{SG}(.)$ is typically included:

$$
\mathcal{L}_\theta = \mathbb{E}_{\boldsymbol{\epsilon},t} \left[ \frac{\tilde{\omega}_t}{\delta^2} \| \boldsymbol{y}_\theta(\boldsymbol{\epsilon}, s) - \text{SG}(\hat{\boldsymbol{y}}_\theta(\boldsymbol{\epsilon}, s)) \|_2^2 \right].
\tag{14}
$$

In our experiments, we also find that it helps use $\tilde{\omega}_t = 1/\lambda'^2_t$ for text-to-image generation.

We can take advantage of higher-order solvers for a more accurate target that reduces the discretization error. For example, one can use Heun's method (Ascher & Petzold, 1998) to first calculate the intermediate value $\tilde{\boldsymbol{y}}_\theta(\boldsymbol{\epsilon}, s)$, and then the final approximation $\hat{\boldsymbol{y}}_\theta(\boldsymbol{\epsilon}, s)$:

$$
\begin{aligned}
\tilde{\boldsymbol{y}}_\theta(\boldsymbol{\epsilon}, s) &= \boldsymbol{y}_\theta(\boldsymbol{\epsilon}, t) + \delta\lambda'_t (\boldsymbol{f}_\phi(\hat{\boldsymbol{x}}_t, t) - \boldsymbol{y}_\theta(\boldsymbol{\epsilon}, t)), \quad \tilde{\boldsymbol{x}}_s = \alpha_s \tilde{\boldsymbol{y}}_\theta(\boldsymbol{\epsilon}, s) + \sigma_s \boldsymbol{\epsilon} \\
\hat{\boldsymbol{y}}_\theta(\boldsymbol{\epsilon}, s) &= \boldsymbol{y}_\theta(\boldsymbol{\epsilon}, t) + \frac{\delta\lambda'_t}{2} [(\boldsymbol{f}_\phi(\hat{\boldsymbol{x}}_t, t) - \boldsymbol{y}_\theta(\boldsymbol{\epsilon}, t)) + (\boldsymbol{f}_\phi(\tilde{\boldsymbol{x}}_s, s) - \tilde{\boldsymbol{y}}_\theta(\boldsymbol{\epsilon}, s))].
\end{aligned}
\tag{15}
$$

Using Heun's method essentially doubles the evaluations of the teacher model during training, while the add-on overheads are manageable as we stop the gradients to the teacher model.

## A.4 Training Algorithm

We summarize the training algorithm of BOOT in Algorithm 1, where by default we assume conditional diffusion model with classifier-free guidance and DDIM solver. Here, for simplicity, we write $\lambda'_t \approx (1 - \frac{\alpha_t \sigma_s}{\sigma_t \alpha_s})/\delta$. For unconditional models, we can simply remove the context sampling part.

# B Connections to Existing Literature

## B.1 Physics Informed Neural Networks (PINNs)

Physics-Informed Neural Networks (PINNs, Raissi et al., 2019) are powerful approaches that combine the strengths of neural networks and physical laws to solve ODEs. Unlike traditional numerical methods, which rely on discretization and iterative solvers, PINNs employ machine learning techniques to approximate the solution of ODEs. The key idea behind PINNs is to incorporate physics-based constraints directly into the training process of neural networks. By embedding the governing equations and available boundary or initial conditions as loss terms, PINNs can effectively learn the underlying physics while simultaneously discovering the solution. This ability makes PINNs highly versatile in solving a wide range of ODEs, including those arising in fluid dynamics, solid mechanics, and other scientific domains. Moreover, PINNs offer several advantages, such as automatic discovery of spatio-temporal patterns and the ability to handle noisy or incomplete data.

Although motivated from different perspectives, BOOT shares similarities with PINNs at a high level, as both aim to learn ODE/PDE solvers directly through neural networks. In the domain of PINNs, solving ODEs can also be simplified into two objectives: the differential equation (DE) loss (Eq. (6)) and the boundary condition (BC) loss (Eq. (8)). The major difference lies in the focus of the two approaches. PINNs primarily focus on learning complex ODEs/PDEs for single problems, where neural networks serve as universal approximators to address the discretization challenges faced by traditional solvers. Moreover, the data space in PINNs is relatively low-dimensional. In contrast, BOOT aims to learn single-step generative models capable of synthesizing data in high-dimensional spaces (e.g., millions of pixels) from random noise inputs and conditions (e.g., labels, prompts). To the best of our knowledge, no existing work has applied similar methods in generative modeling. Additionally, while standard PINNs typically compute derivatives (Eq. (6)) directly using auto-differentiation, in this paper, we employ the finite difference method and propose a bootstrapping-based algorithm.

## B.2 Consistency Models / TRACT

The most related previous works to our research are Consistency Models (Song et al., 2023) and concurrently TRACT (Berthelot et al., 2023), which propose bootstrapping-style algorithms for distilling diffusion models. These approaches map an intermediate noisy training example at time step $t$ to the teacher's $t$-step denoising outputs using the DDIM inference procedure. The training target for the student is constructed by running the teacher model with one step, followed by the self-teacher with $t-1$ steps. As illustrated in Fig. 11, BOOT takes a different approach to bootstrapping. It starts from the Gaussian noise prior and directly maps it to an intermediate step $t$ in one shot. This change has significant modeling implications, as it does not require any training data and can achieve data-free distillation, a capability that none of the prior works possess.

## B.3 Single-step Generative Models

BOOT is also related to other single-step generative models, including VAEs (Kingma & Welling, 2013) and GANs (Goodfellow et al., 2014b), which aim to synthesize data in a single forward pass. However, BOOT does not require an encoder network like VAEs. Thanks to the power of the underlying diffusion model, BOOT can produce higher-contrast and more realistic samples. In comparison to GANs, BOOT does not require a discriminator or critic network. Furthermore, the distillation process of BOOT enables better-controlled exploration of the text-image joint space, which is explored by the pretrained diffusion models, resulting in more coherent and realistic samples in text-guided generation. Additionally, BOOT is more stable to learn compared to GANs, which are challenging to train due to the adversarial nature of maintaining a balance between the generator and discriminator networks.

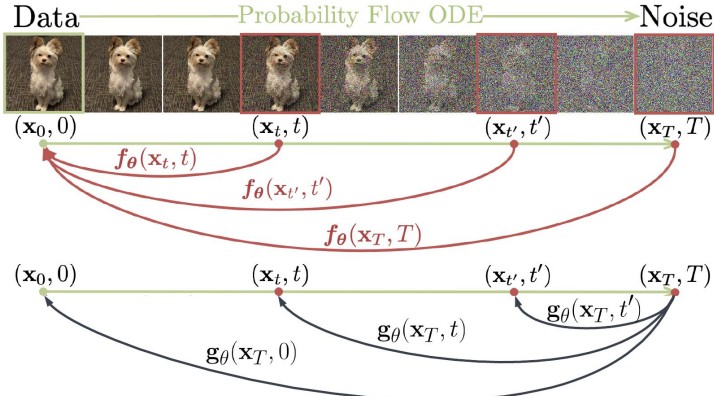

Figure 11: Comparison of Consistency Model (Song et al., 2023) (red ↑) and BOOT (black ↓) highlighting the opposing prediction pathways.

## C  ADDITIONAL EXPERIMENTAL SETTINGS

### C.1  DATASETS

While the proposed method is data-free, we list the additional dataset information that used to train our teacher diffusion models:

**FFHQ** (https://github.com/NVlabs/ffhq-dataset) contains 70k images of real human faces in resolution of $1024 \times 1024$. In most of our experiments, we resize the images to a low resolution at $64 \times 64$ for early-stage benchmarking.

**LSUN** (https://www.yf.io/p/lsun) is a collection of large-scale image dataset containing 10 scenes and 20 object categories. Following previous works (Song et al., 2023), we choose the category Bedroom (3M images), and train an unconditional diffusion teacher. All images are resized to $256 \times 256$ with center-crop. We use LSUN to validate the ability of learning in relative high-resolution scenarios.

**ImageNet-1K** (https://image-net.org/download.php) contains 1.28M images across 1000 classes. We directly merge all the training images with class labels and train a class-conditioned diffusion teacher. All images are resized to $64 \times 64$ with center-crop. To support test-time classifier-free guidance, the teacher model is trained with 0.2 unconditional probability.

As we do not need to train our own teacher models for text-to-image experiments, no additional text-image pairs are required in this paper. However, our distillation still requires the text conditions for querying the teacher diffusion. To better capture and generalize the real user preference of such diffusion models, we choose to adopt the collected prompt datasets:

**DiffusionDB** (https://poloclub.github.io/diffusiondb/) contains 14M images generated by Stable Diffusion using prompts and hyperparameters specified by users. For the purpose of our experiments, we only keep the text prompts and discard all model-generated images as well as meta-data and hyperparameters so that they can be used for different teacher models. We use the same prompts for both latent and pixel space models.

### C.2  TEXT-TO-IMAGE TEACHERS

We directly choose the recently open-sourced large-scale diffusion models as our teacher models. More specifically, we looked into the following models:

**StableDiffusion (SD)** (https://github.com/Stability-AI/stablediffusion) is an open-source text-to-image latent diffusion model (Rombach et al., 2021) conditioned on the penultimate text embeddings of a CLIP ViT-H/14 (Radford et al., 2021) text encoder. Different standard diffusion models, SD performs diffusion purely in the latent space. In this work, we use the checkpoint of **SD v2.1-Base** (https://huggingface.co/stabilityai/stable-diffusion-2-1-base) as our teacher which first generates in $64 \times 64$ latent space,

| Hyperparameter | Image Generation | | | Text-to-Image | | |
|---|---|---|---|---|---|---|
| | FFHQ | LSUN | ImageNet | SD-Base | IF-I-L | IF-II-M |
| **Architecture** | | | | | | |
| Denosing resolution | $64 \times 64$ | $256 \times 256$ | $64 \times 64$ | $64 \times 64$ | $64 \times 64$ | $256 \times 256$ |
| Base channels | 128 | 128 | 192 | | | |
| Multipliers | 1,2,3,4 | 1,1,2,2,4,4 | 1,2,3,4 | | | |
| # of Resblocks | 1 | 1 | 2 | | | |
| Attention resolutions | 8,16 | 8,16 | 8,16 | | – Default – | |
| Noise schedule | cosine | cosine | cosine | | | |
| Model Prediction | signal | signal | signal | | | |
| Text Encoder | - | - | - | CLIP | T5 | T5 |
| **Training** | | | | | | |
| Loss weighting | uniform | uniform | uniform | $\lambda_t'^{-2}$ | $\lambda_t'^{-2}$ | $\lambda_t'^{-2}$ |
| Bootstrapping step size | 0.04 | 0.04 | 0.04 | 0.01 | 0.04 | 0.04 |
| CFG weight | - | - | $1 \sim 5$ | 7.5 | 7.0 | 4.0 |
| Learning rate | 1e-4 | 1e-4 | 3e-4 | 2e-5 | 2e-5 | 2e-5 |
| Batch size | 128 | 128 | 1024 | 64 | 64 | 32 |
| EMA decay rate | 0.9999 | 0.9999 | 0.9999 | 0.9999 | 0.9999 | 0.9999 |
| Training iterations | 500k | 500k | 300k | 500k | 500k | 100k |

Table 3: Hyperparameters used for training BOOT. The CFG weights for text-to-image models are determined based on the default value of the open-source codebase.

and then directly upscaled to $512 \times 512$ resolution with the pre-trained VAE decoder. The teacher model was trained on subsets of LAION-5B (Schuhmann et al., 2022) with noise prediction objective.

**DeepFloyd IF (IF)** (`https://github.com/deep-floyd/IF`) is a recently open-source text-to-image model with a high degree of photorealism and language understanding. IF is a modular composed of a frozen text encoder and three cascaded pixel diffusion modules, similar to Imagen (Saharia et al., 2022): a base model that generates $64 \times 64$ image based on text prompt and two super-resolution models ($256 \times 256$, $1024 \times 1024$). All stages of the model utilize a frozen text encoder based on the T5 (Raffel et al., 2020) to extract text embeddings, which are then fed into a UNet architecture enhanced with cross-attention and attention pooling. Models were trained on 1.2B text-image pairs (based on LAION (Schuhmann et al., 2022) and few additional internal datasets) with noise prediction objective. In this paper, we conduct experiments on the first two resolutions ($64 \times 64$, $256 \times 256$) with the checkpoints of **IF-I-L-v1.0** (`https://huggingface.co/DeepFloyd/IF-I-L-v1.0`) and **IF-II-M-v1.0** (`https://huggingface.co/DeepFloyd/IF-II-M-v1.0`).

## C.3  MODEL ARCHITECTURES

We follow the standard U-Net architecture (Nichol & Dhariwal, 2021) for image generation benchmarks and adopt the hyperparameters similar in f-DM (Gu et al., 2022). For text-to-image applications, we keep the default architecture setups from the teacher models unchanged. As mentioned in the main paper, we initialize the weights of the student models directly from the pretrained checkpoints and use *zero* initialization for the newly added modules, such as target time and CFG weight embeddings. We include additional architecture details in the Table 3.

## C.4  TRAINING DETAILS

All models for all the tasks are trained on the same resources of 8 NVIDIA A100 GPUs for 500K updates. Training roughly takes $3 \sim 7$ days to converge depending on the model sizes. We train all our models with the AdamW (Loshchilov & Hutter, 2017) optimizer, with no learning rate decay or warm-up, and no weight decay. Standard EMA to the weights is also applied for student models. Since our methods are data-free, there is no additional overhead on data storage and loading except for the text prompts, which are much smaller and can be efficiently loaded into memory.

Learning the boundary loss requires additional NFEs during each training step. In practice, we apply the boundary loss less frequently (e.g., computing the boundary condition every 4 iterations and

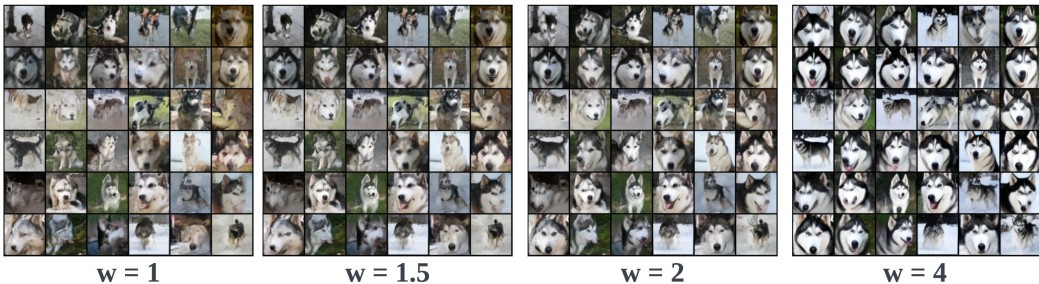

| w = 1 | w = 1.5 | w = 2 | w = 4 |

Figure 12: The distilled student is able to trade generation quality with diversity based on CFG weights.

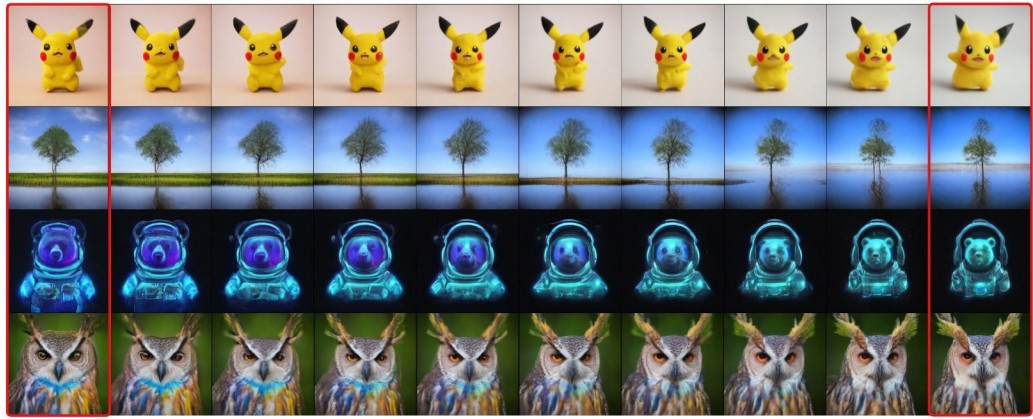

Figure 13: Latent space interpolation of the student model distilled from the IF teacher. We randomly sample two noises to generate images (shown in red boxes) given the same text prompts, and then linearly interpolate the noises to synthesize images shown in the middle.

setting the loss to be 0 otherwise) to improve the overall training efficiency. Note that distilling from the class-conditioned / text-to-image teachers requires multiple forward passes due to CFG, which relatively slows down the training compared to unconditional models.

Distilling from the DeepFloyd IF teacher requires learning from two stages. In this paper, we can easily achieve that by first distilling the first-stage model into single-step with BOOT, and then distilling the upscaler model based on the output of the first-stage student. Following the original paper (Saharia et al., 2022), noise augmentation is also applied on the first-stage output where we set the noise-level as 250 *. For more training hyperparameters, please refer to Table 3.

# D    ADDITIONAL SAMPLES FROM BOOT

Finally, we provide additional qualitative comparisons for the unconditional models of CIFAR-10 $32 \times 32$ (Fig. 14), FFHQ $64 \times 64$ (Fig. 15), LSUN $256 \times 256$ (Fig. 16), the class-conditional model of ImageNet $64 \times 64$ (Fig. 17), and comparisons for text-to-image generation based on DeepFloyd-IF ($64 \times 64$ in Figs. 18 and 21, $256 \times 256$ in Figs. 1 and 8 to 10) and StableDiffusion ($512 \times 512$ in Figs. 20 and 22).

---

*https://github.com/huggingface/diffusers/blob/main/src/diffusers/pipelines/deepfloyd_if/pipeline_if_superresolution.py#L715

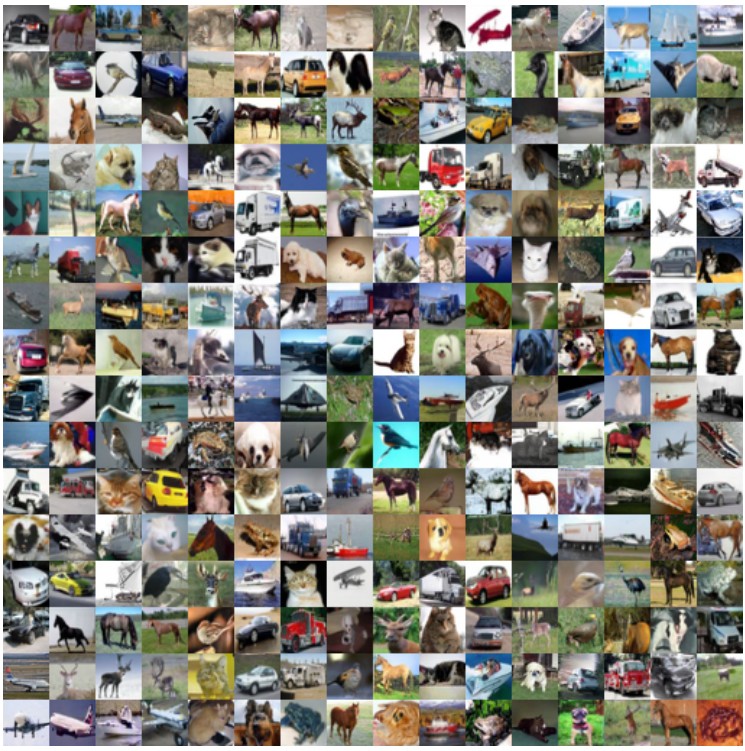

(a) Random samples from EDM teacher.

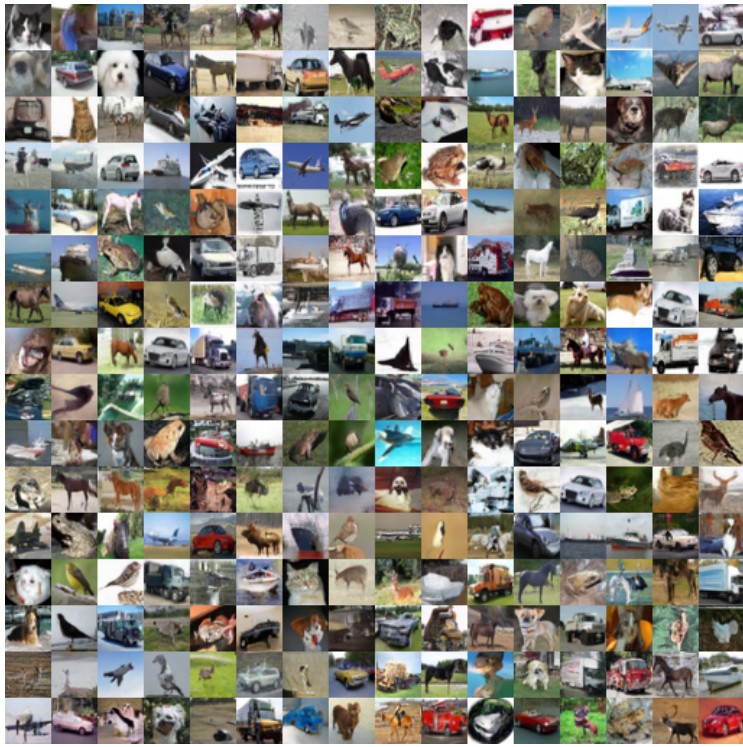

(b) Random samples from BOOT student.

Figure 14: Qualitative comparison between EDM teacher and BOOT samples on CIFAR-10 datasets.

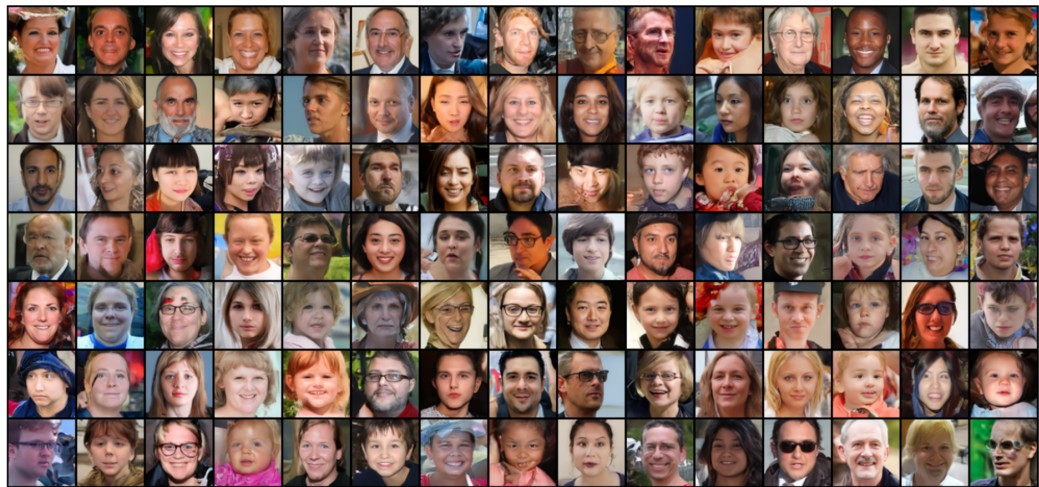

(a) Diffusion Model Teacher (50 steps / 1.2 FPS )

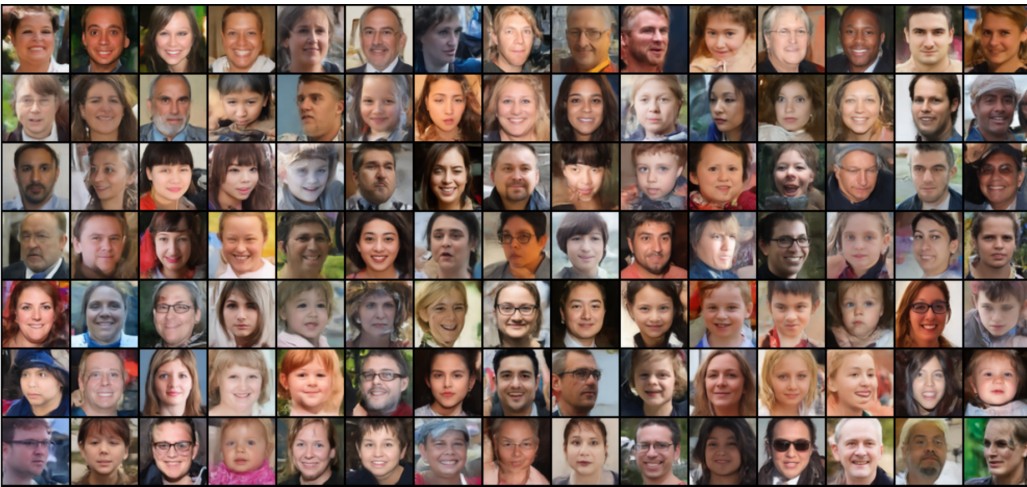

(b) Distilled Student with BOOT (1 step / 54 FPS )

Figure 15: Uncurated samples from FFHQ $64 \times 64$. All corresponding samples use the same initial noise for the DDIM teacher and the single-step BOOT student.

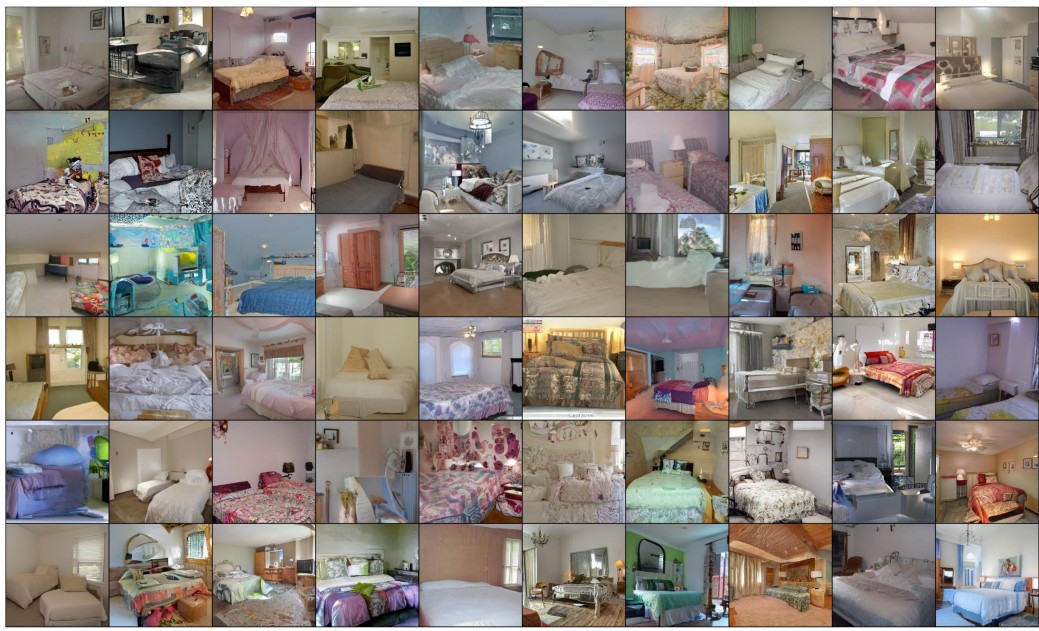

(a) Diffusion Model Teacher (50 steps / 0.6 FPS )

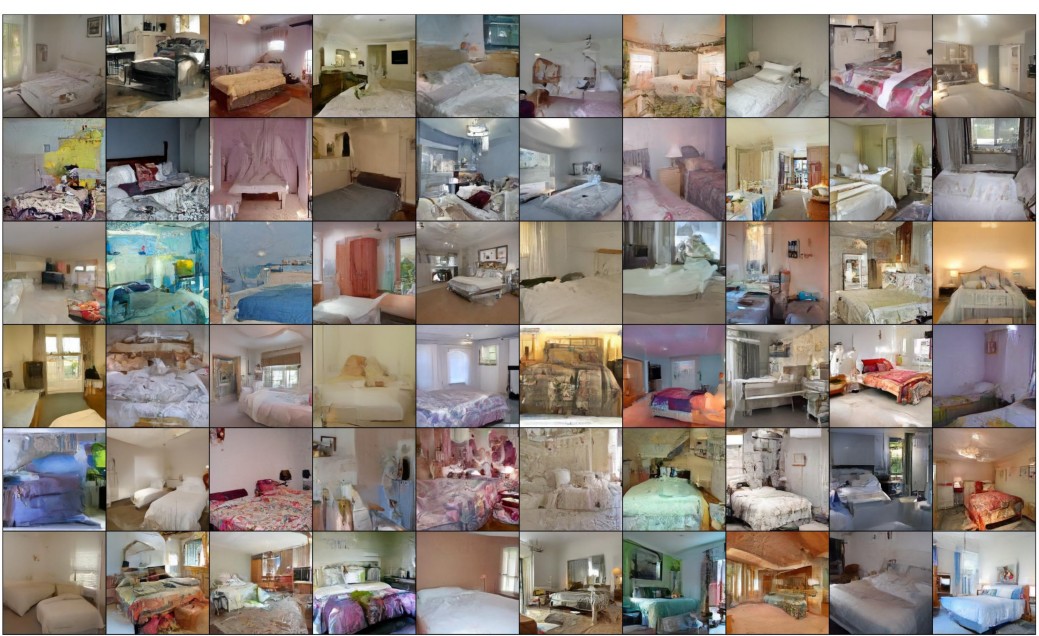

(b) Distilled Student with BOOT (1 step / 32 FPS )

Figure 16: Uncurated samples from LSUN Bedroom $256 \times 256$. All corresponding samples use the same initial noise for the DDIM teacher and the single-step BOOT student.

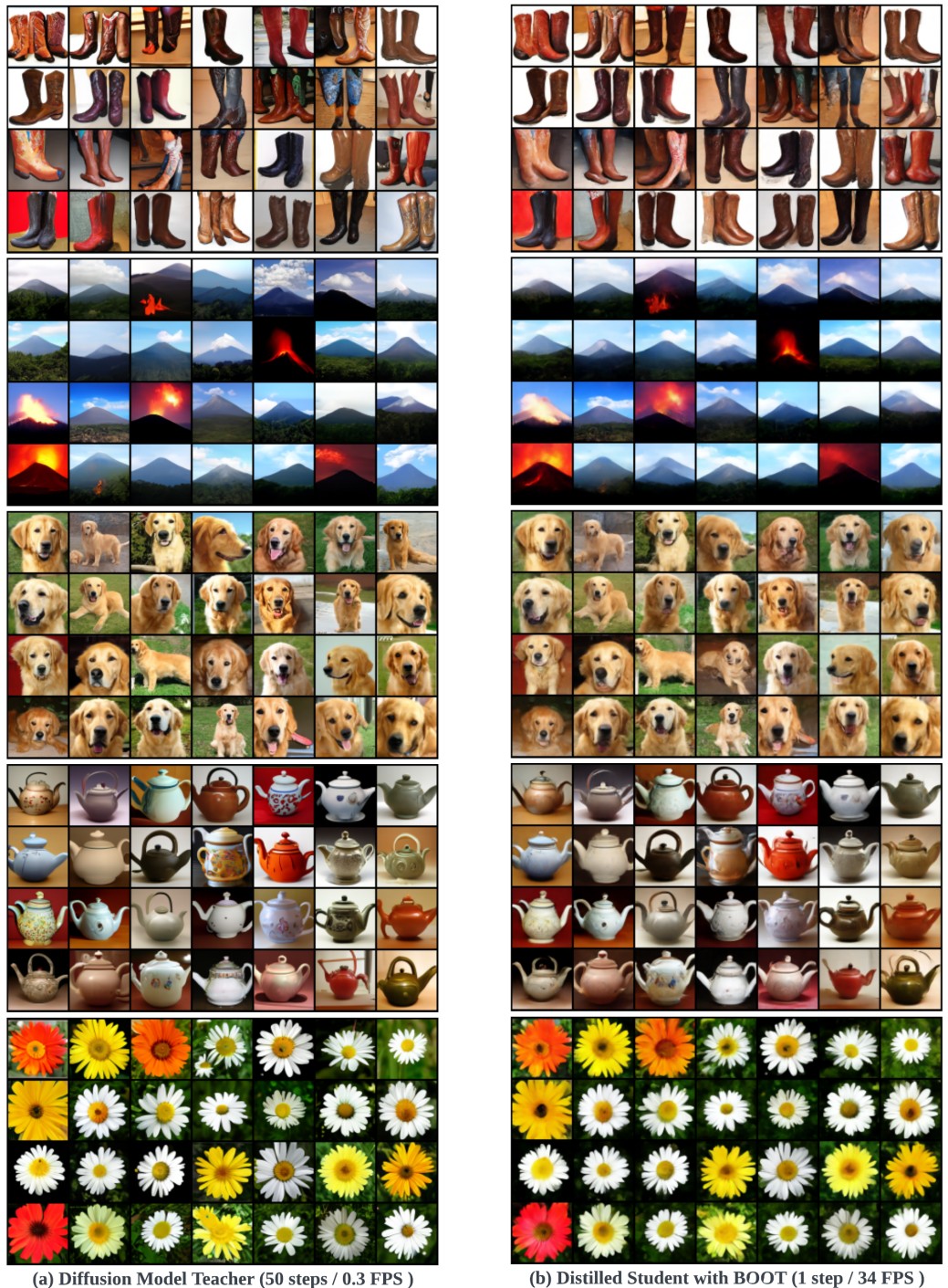

**(a) Diffusion Model Teacher (50 steps / 0.3 FPS )**       **(b) Distilled Student with BOOT (1 step / 34 FPS )**

Figure 17: Uncurated class-conditioned samples from ImageNet $64 \times 64$. All corresponding samples use the same initial noise for the DDIM teacher and the single-step BOOT student. Classes from top to bottom: *cowboy boot, volcano, golden retriever, teapot, daisy*. The diffusion model uses CFG with $w = 3$, and our student model conditions on the same weight.

**Prompt:** *portrait of Einstein, award winning art – style art, detailed*

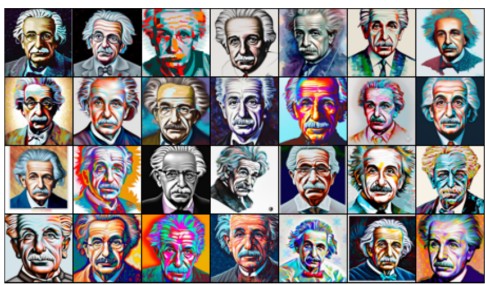 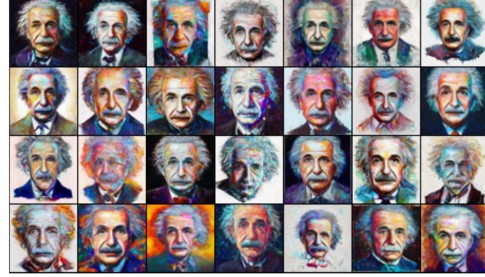

**Prompt:** *grand prismatic springs in a coffee cup sitting on kitchen table*

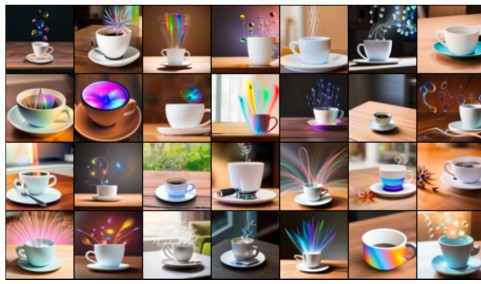 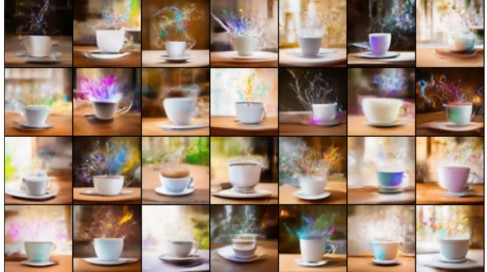

**Prompt:** *A raccoon wearing formal clothes, wearing a tophat. Oil painting in the style of Rembrandt*

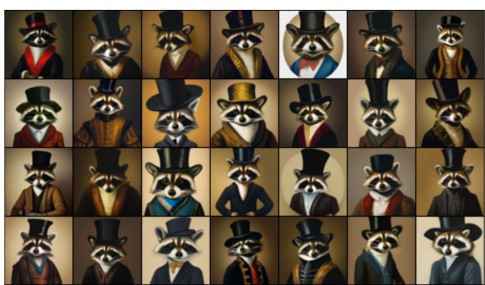 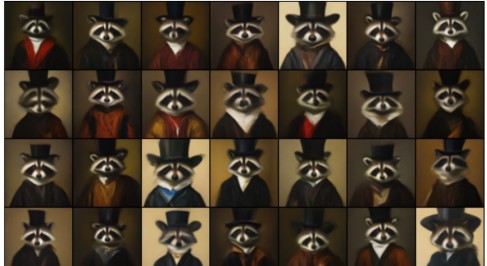

**Prompt:** *A photorealistic illustration of Darth Vader in the style of Van Gogh starry night painting.*

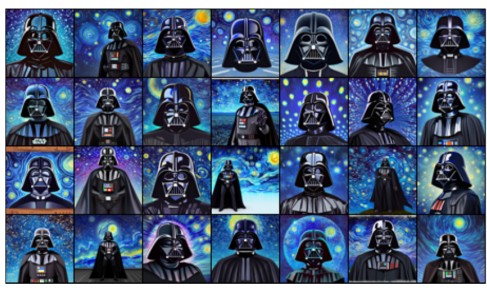 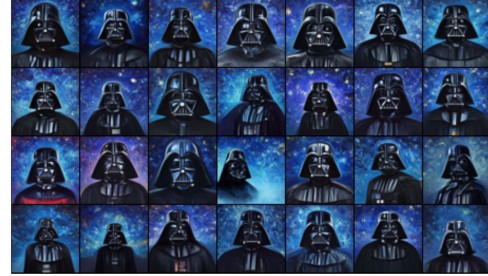

(a) Diffusion Model Teacher (50 steps)   (b) Distilled Student with BOOT (1 step)

Figure 18: Uncurated text-conditioned image generation distilled from DeepFloyd IF (the first stage model, images are at $64 \times 64$). All corresponding samples use the same initial noise for the DDIM teacher and the single-step BOOT student. The specific prompts are shown above the images.

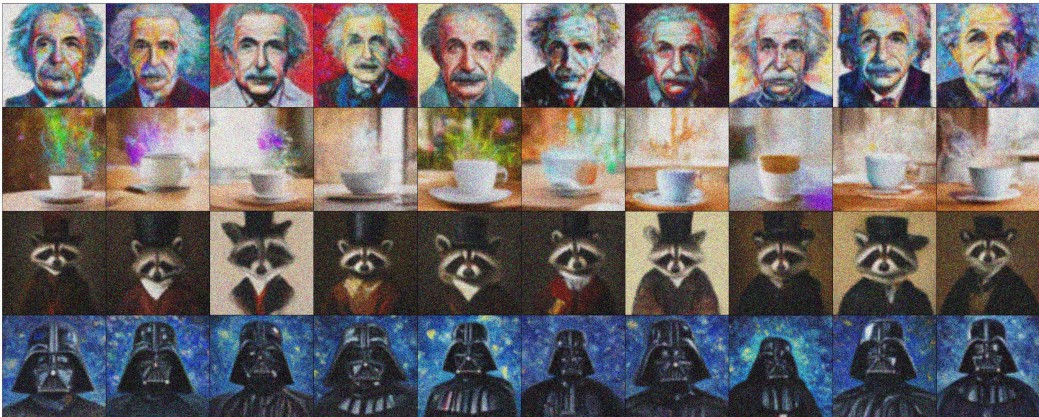

(a) First-stage Distilled Student Output + Bilinear Upsample + Noise Augment

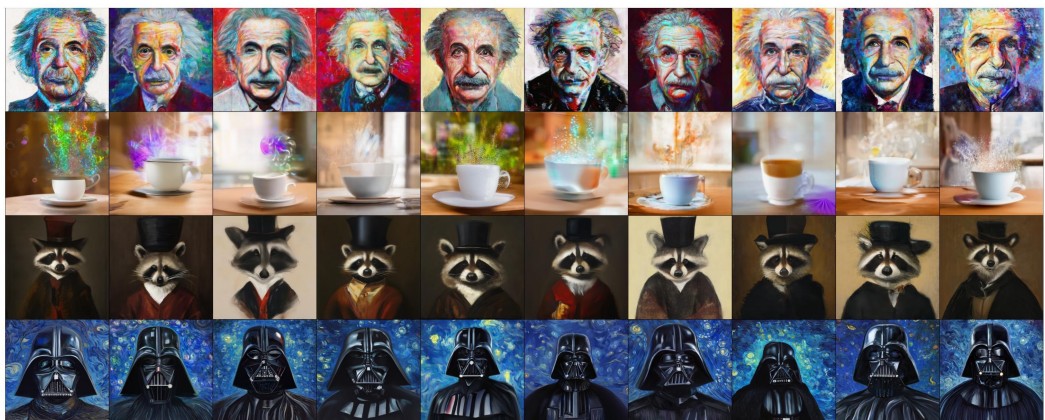

(b) Diffusion Model Teacher (50 Steps + CFG)

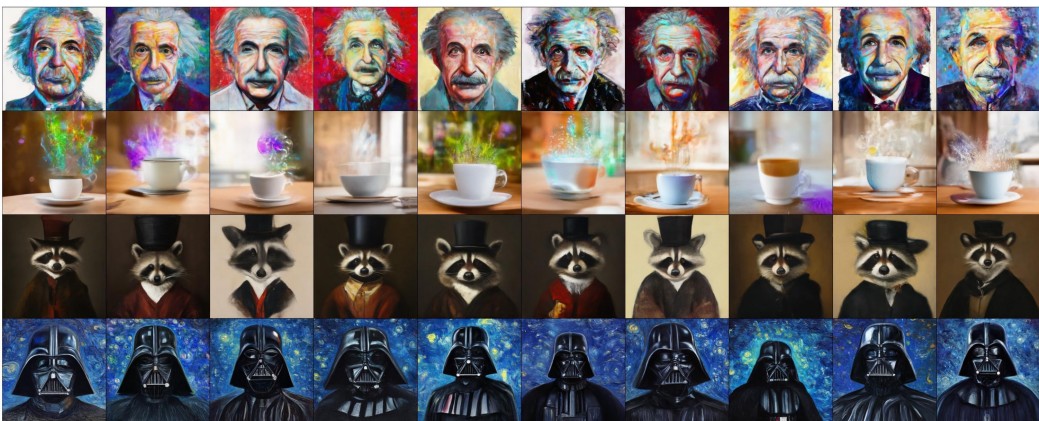

(c) Distilled Student with BOOT (1 Step)

Figure 19: Given the $64 \times 64$ outputs from Fig. 18, we also show comparison for the second-stage models which upscale the images to $256 \times 256$. All corresponding samples use the same initial noise for the DDIM teacher and the single-step BOOT student.

**Prompt:** *portrait of Einstein, award winning art – style art, detailed*

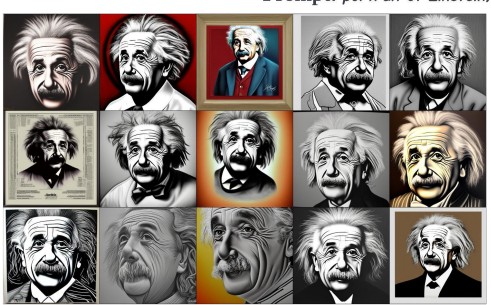 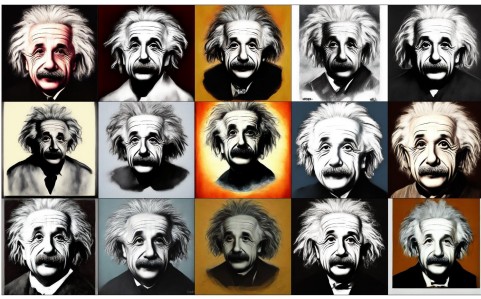

**Prompt:** *grand prismatic springs in a coffee cup sitting on kitchen table*

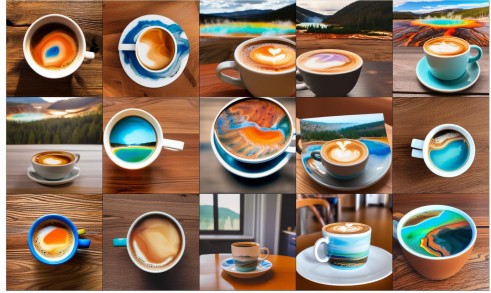 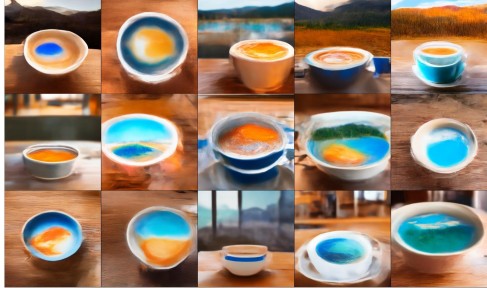

**Prompt:** *A raccoon wearing formal clothes, wearing a tophat. Oil painting in the style of Rembrandt*

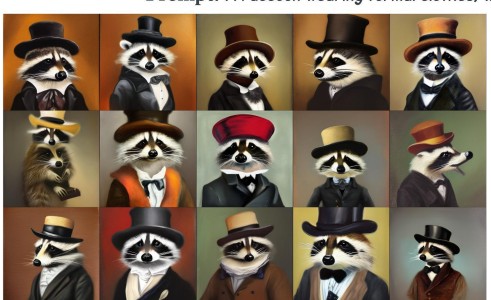 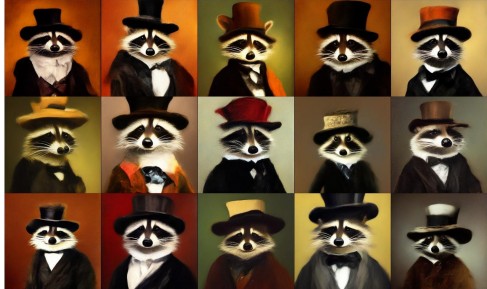

**Prompt:** *A photorealistic illustration of Darth Vader in the style of Van Gogh starry night painting.*

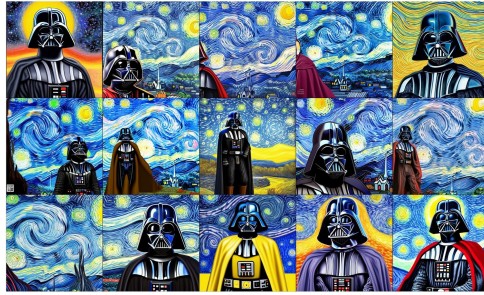 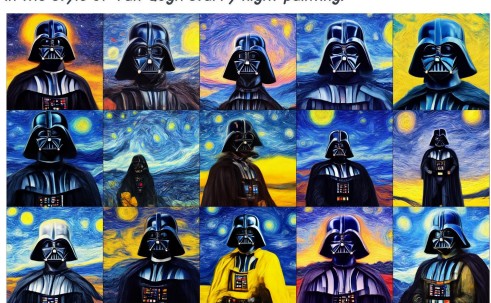

(a) Diffusion Model Teacher (50 steps)          (b) Distilled Student with BOOT (1 step)

Figure 20: Uncurated text-conditioned image generation distilled from StableDiffusion (latent diffusion in $64 \times 64$, images are upscaled to $512 \times 512$ with the pre-trained VAE decoder). All corresponding samples use the same initial noise for the DDIM teacher and the single-step BOOT student. We use the same prompts as in Fig. 18 for better comparison.

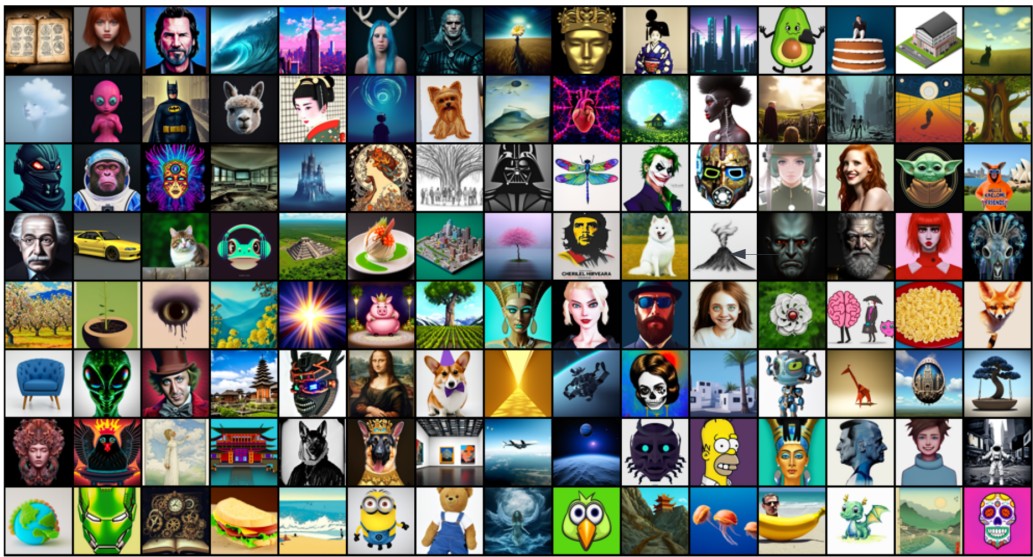

(a) Random Samples from Diffusion Model Teacher (50 steps with CFG)

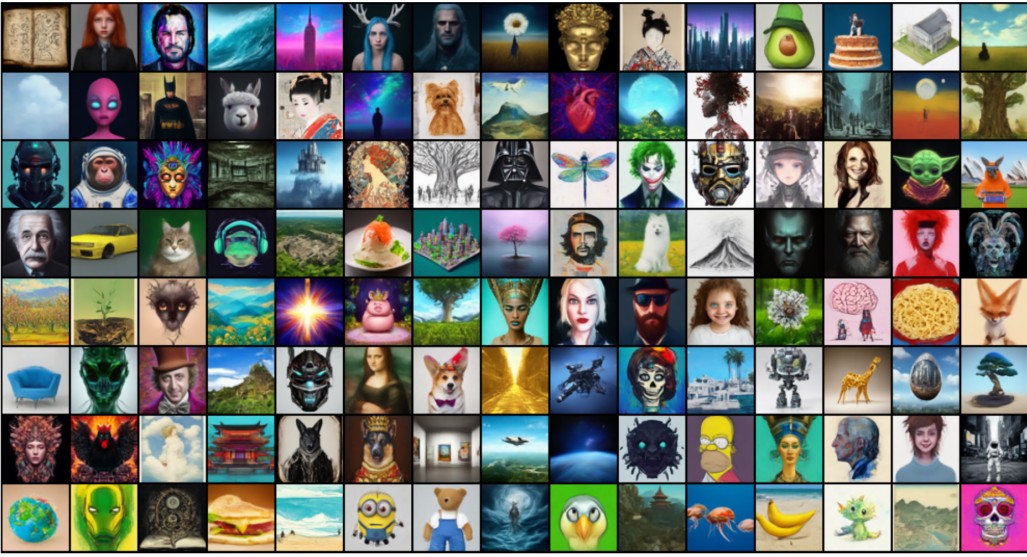

(b) Random Samples from Distilled Student with BOOT (1 step)

Figure 21: Uncurated text-conditioned image generation distilled from DeepFloyd IF (the first stage model, images are at $64 \times 64$) given sampled text prompts from *diffusiondb* (Wang et al., 2022) randomly. All corresponding samples use the same initial noise for the DDIM teacher and the single-step student. Besides, we also show examples from the distilled model at $256 \times 256$ in Fig. 1.

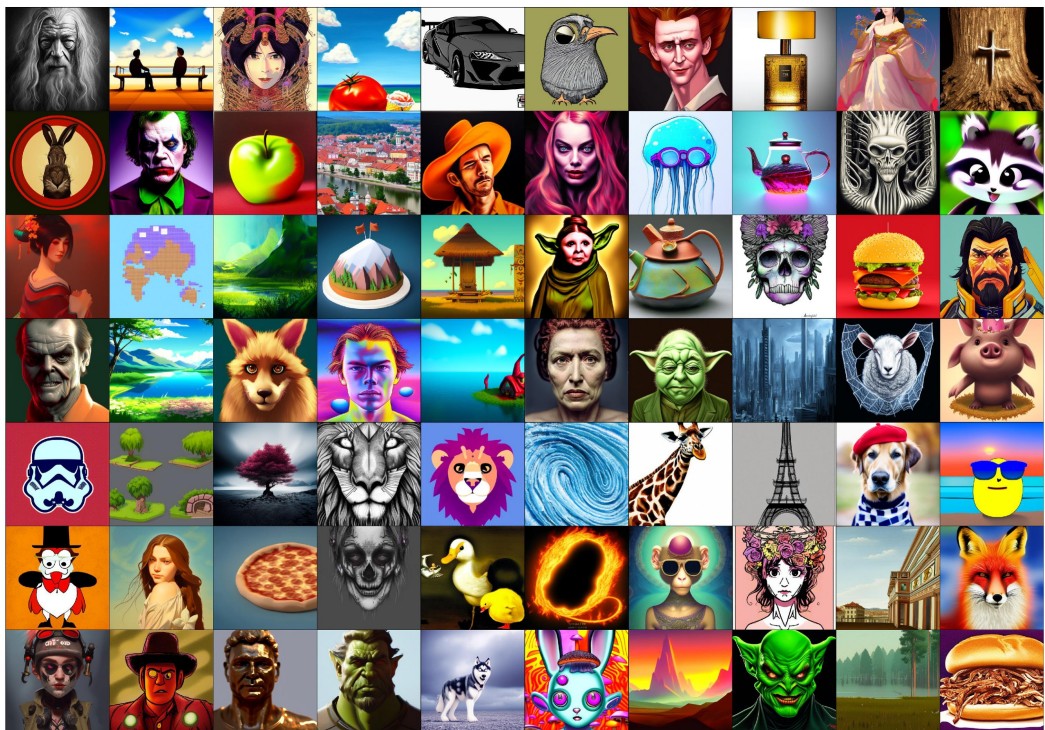

(a) Random Samples from Diffusion Model Teacher (50 steps with CFG)

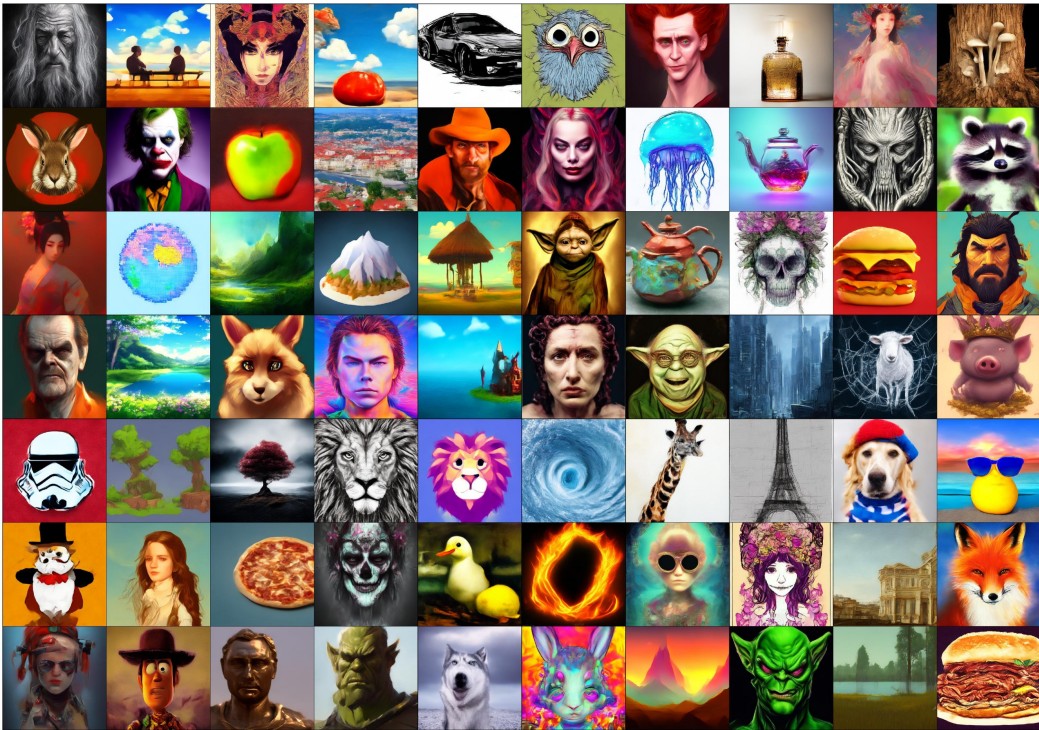

(b) Random Samples from Distilled Student with BOOT (1 step)

Figure 22: Uncurated text-conditioned image generation distilled from StableDiffusion (latent diffusion in $64 \times 64$, images are upscaled to $512 \times 512$ with the pre-trained VAE decoder) given sampled text prompts from *diffusiondb* (Wang et al., 2022) randomly. All corresponding samples use the same initial noise for the DDIM teacher and the single-step student.

