# OpenReview forum: "BOOT: Data-free Distillation of Denoising Diffusion Models with Bootstrapping"
_ICLR.cc/2024/Conference — ICLR 2024 Conference Withdrawn Submission_

### Official Review · Reviewer_Fzoy · 2023-10-22

**Soundness:** 3 good
**Presentation:** 3 good
**Contribution:** 3 good
**Rating:** 6
**Confidence:** 4

**Summary:**

The paper proposes a novel distillation technique for denoising diffusion models with the specific goal of single-step synthesis. Conceptually, the work is closely related to Consistency Models, but instead of predicting clean data $x$ from anywhere along the probability flow ODE trajectory, BOOT instead trains to predict any $x_t$ along the trajectory from the starting noise $\epsilon$, using a bootstrapping approach. At convergence, the model can directly predict $x_0$. To this end, the paper develops a novel training objective based on the Signal-ODE, somewhat similar to consistency matching, leveraging bootstrapping. A crucial advantage of the method is that it does not require the diffusion model's training dataset. Instead, any diffusion model can be directly distilled without access to the model's training dataset. This is in contrast to previous distillation methods. Experimentally, the method is evaluated on several standard image datasets, overall demonstrating promising performance. Importantly, the paper also demonstrates distillation of large text-to-image models with good results.

**Strengths:**

There are several strengths:
- *Significance:* The method's performance is almost as good as Consistency Models' performance, while not requiring training data. This makes the method very practical and useful. The fact that BOOT is data-free is the main strength of the paper.
- *Clarity:* The paper is overall well written and easy to follow.
- *Originality and Novelty:* The method is novel and well-motivated (although conceptually closely related to previous works like Consistency Models).
- The qualitative experiments on large text-to-image models are strong. Distilling such large models is usually difficult, because previous distillation method would require access to the large training data set.

**Weaknesses:**

The are also some weaknesses:
- Performance-wise, the method still does not perform as well as Consistency Models or TRACT on CIFAR10.
- Conceptually, the method is novel, but incremental compared to Consistency Models or TRACT. While these papers learn to predict $x_0$ from any $x_t$ using a consistency objective, BOOT learns predict any $x_t$ from $\epsilon$ using bootstrapping with a very related objective.
- Multistep sampling is not possible with BOOT. It is strict one-step generation, as opposed to previous distillation methods.
- The paper has experiments on FFHQ, LSUN and ImageNet, but only compares to regular DDIM and DDPM samplers here. Quantitative comparisons to previous distillation methods are only done for CIFAR10. However, TRACT, Progressive Distillation and Consistency Models all have quantitative results for ImageNet64. These results were not shown in the paper, although BOOT does evaluate on ImageNet. However, it does seem to achieve worse performance than these baselines. I am also wondering why TRACT was not even included in the CIFAR10 comparisons (TRACT is discussed and referenced in the text, after all).
- BOOT's recall scores in Table 1 are generally quite a bit lower than those of regular DDPM or DDIM sampling (with many steps). This means that some sample diversity and mode coverage is lost in the BOOT distillation.

*Conclusion:* Overall, I think that the paper is on the one hand a bit incremental and the raw performance is not better than previous approaches. As discussed above, there are some weaknesses and concerns. However, the data-free nature is very valuable and represents an important innovation towards practical fast sampling with distillation methods for diffusion models, which is very important. Therefore, I am leaning towards recommending acceptance.

**Questions:**

- In section 3.2, the authors write that the "incremental improvement term in the training target is mostly non-zero". I can imagine this to be true, but "mostly non-zero" seems vague. Could the authors quantify this and maybe analyze and demonstrate that this term indeed does never collapse to zero?
- How is $\delta$ chosen? How robust is the method with respect to different $\delta$? It's an important and new parameter. An ablation experiment could be interesting.

I also have a few suggestions:
- NFE has been used as abbreviation "Neural Function Evaluation" (in introduction), or "Number of Function Evaluations" (in 2.1). It should only be used in one way, consistently.
- When discussing Consistency Models in the Background section, the paper does not mention the boundary conditions of Consistency Models at $x_0$. I would recommend to add this. This is crucial to them, and somewhat analogous and related to the boundary conditions which BOOT has at $x_T$.
- The Signal-ODE is similar to the DDIM-ODE, which also models the signal (in contrast to the PF-ODE). DDIM-ODE or Signal-ODE are reparametrized versions of the original PF-ODE. I would suggest the authors to make this clear. For the DDIM-ODE and how it is related to the Signal-ODE, see, for instance, Eq. (8) in Progressive Distillation paper (division by $\alpha_s$ and reparametrizing $z_s/\alpha_s \rightarrow \hat{z}_s$ accordingly yields exactly the Signal-ODE; also see 4.3 of the DDIM paper).

---

### Official Review · Reviewer_aBRM · 2023-10-28

**Soundness:** 2 fair
**Presentation:** 3 good
**Contribution:** 2 fair
**Rating:** 3
**Confidence:** 4

**Summary:**

The paper introduces a novel data-free distillation method for diffusion models. The proposed approach does not require expensive simulations at the training stage. This makes the training much more efficient compared to alternative data-free distillation techniques.

**Strengths:**

* The proposed method is interesting and novel. I especially like that it represents an opposite integrator learning perspective compared to consistency models.
* The paper is clearly written and well-presented.

**Weaknesses:**

### Evaluation
* Significantly worse performance compared to CM[1] on all datasets, especially on the ones in Table 1. Please consider providing the results with LPIPS loss on the datasets from Table 1 to better understand the gap with CM.
* No comparison with TRACT[2] and Diff-Instruct[3] and GANs.
* In Table 1, no comparison with single and low-step baselines at all, please add the baselines from Table 2 + [2,3] + GANs. DDIM and DDPM are also suboptimal in the high NFE setting, please add EDM[4] and ADM[5].
* For text-conditional generation, only CLIP-score is provided. FID score should be reported as well. Human preference evaluation would also be highly beneficial.

### Motivation
* The importance of data-free methods seems a bit overclaimed. I believe that "data-free" does not justify the drastic drop in the generation performance.
* I disagree that “text-to-image generation models **require** billions of paired data for training.” Billion-scale datasets like LAION usually contain a lot of low quality pairs. Thus, the properly filtered data with millions of pairs may result in comparable or superior t2i generation [6].
* I also cannot agree with the claim: “One possible solution is to use a different dataset for distillation; however, the mismatch in the distributions of the two datasets would result in suboptimal distillation performance.” In my opinion, it is still an open question: how many training samples and what datasets are really needed for effective distillation.

[1] Song et al., Consistency models.

[2] Berthelot et al., TRACT: Denoising Diffusion Models with Transitive Closure Time-Distillation.

[3] Luo et al., Diff-Instruct: A Universal Approach for Transferring Knowledge From Pre-trained Diffusion Models.

[4] Karras et al., Elucidating the Design Space of Diffusion-Based Generative Models.

[5] Dhariwal and Alex Nichol, Diffusion Models Beat GANs on Image Synthesis.

[6] Chen et al., PixArt-α: Fast Training of Diffusion Transformer for Photorealistic Text-to-Image Synthesis

**Questions:**

* Please address the weaknesses.
* The authors trained the signal prediction teacher models for Imagenet, LSUN-bedroom and FFHQ. Why is it a problem to consider publicly available noise-prediction models and obtain the signal prediction as $(\hat{x}_t - \sigma_t\cdot\epsilon)/\alpha_t$?
* I found a bit confusing "$y_T=\epsilon{\sim}N(0, I)$" in the end of Section 3.1. From my perspective, $y_T$ is supposed to be the denoised prediction at a time step $T$. Can you elaborate please why $y_T{\sim}N(0, I)$ is a reasonable initialization for the Signal-ODE?
* In Sec 2.1, please add that $f_\psi$ is a denoised output.

---

### Official Review · Reviewer_Xhbi · 2023-10-31

**Soundness:** 2 fair
**Presentation:** 2 fair
**Contribution:** 2 fair
**Rating:** 3
**Confidence:** 4

**Summary:**

This paper introduces a method for diffusion distillation, drawing parallels with consistency models. The proposed technique involves training a generator to take complete Gaussian noise as input and output noisy image corresponding to various possible timestep in common diffusion models. Recognizing the challenge for predicting noisy image with neural networks, the authors requires the network to predict a low-frequency signal image for each timestep. To train the generator, the authors propose a loss to match these predicted images across consecutive timesteps, similar to that used in Consistency Models (CM). Additionally, the paper presents techniques to ensure boundary conditions are adhered to, aligning with the output of diffusion models when the input is entirely noise.

The efficacy of the approach is validated through a series of experiments on multiple benchmarks.

**Strengths:**

S1: The method proposed is novel, and the authors have conducted a comprehensive set of experiments across various benchmarks to test its efficacy.

S2: The technical aspects introduced—namely the handling of noisy targets, the enforcement of boundary conditions, and the finite difference approximation for time derivatives—constitute the paper's core contributions, which could be of interest to those working on diffusion models.

S3: The paper presents a range of text-to-image generation results that scale beyond previous works.

**Weaknesses:**

W1: The paper claims novelty in eliminating the need for real data through its method. However, this advantage is not exclusively unique to the proposed approach since direct distillation methods can achieve similar results. With the advancement of ODE solvers capable of efficiently sampling in 10 to 15 steps, the computational cost of constructing such dataset required for distillation may not be as prohibitive as suggested [4], reducing the perceived novelty of this feature.

W2: The results presented do not clearly outperform established baselines. On CIFAR-10, the model's performance is notably inferior. Additionally, the compared direct distillation style methods [1, 2] are using worse network architectures, so that the advantages of the proposed method may not be evident. The absence of a distillation baseline comparison in ImageNet experiments (and seemingly weaker performance relative to consistency models and DFNO [3]) raise questions about the method's effectiveness

W3: Alongside performance concerns, it's uncertain whether the new formulation offers any technical superiority over consistency models. For instance, the requirement to enforce boundary conditions adds complexity, whereas consistency models naturally incorporate these conditions without additional overhead.


[1] Luhman, Eric, and Troy Luhman. "Knowledge distillation in iterative generative models for improved sampling speed." arXiv preprint arXiv:2101.02388 (2021).

[2] Liu, Xingchao, Chengyue Gong, and Qiang Liu. "Flow straight and fast: Learning to generate and transfer data with rectified flow." arXiv preprint arXiv:2209.03003 (2022).

[3] Zheng, Hongkai, et al. "Fast sampling of diffusion models via operator learning." International Conference on Machine Learning. PMLR, 2023.

[4] Liu, Xingchao, et al. "Instaflow: One step is enough for high-quality diffusion-based text-to-image generation." arXiv preprint arXiv:2309.06380 (2023).
APA

**Questions:**

Please find comments in the weakness section. Additionally, for COCO results, could you also report the FID (currently only CLIP score is presented) and compare with baselines?